# Stretching the skin immediately enhances perceived stiffness and gradually enhances the predictive control of grip force

Mor Farajian[1,2], Raz Leib[1,2], Hanna Kossowsky[1], Tomer Zaidenberg[1], Ferdinando A Mussa-Ivaldi[3,4], Ilana Nisky[1,2]*

[1]Department of Biomedical Engineering, Ben-Gurion University of the Negev, Be'er Sheva, Israel; [2]Zlotowski Center for Neuroscience, Ben-Gurion University of the Negev, Be'er Sheva, Israel; [3]Shirley Ryan AbilityLab, Chicago, United States; [4]Department of Biomedical Engineering, Northwestern University, Evanston, United States

**Abstract** When manipulating objects, we use kinesthetic and tactile information to form an internal representation of their mechanical properties for cognitive perception and for preventing their slippage using predictive control of grip force. A major challenge in understanding the dissociable contributions of tactile and kinesthetic information to perception and action is the natural coupling between them. Unlike previous studies that addressed this question either by focusing on impaired sensory processing in patients or using local anesthesia, we used a behavioral study with a programmable mechatronic device that stretches the skin of the fingertips to address this issue in the intact sensorimotor system. We found that artificial skin-stretch increases the predictive grip force modulation in anticipation of the load force. Moreover, the stretch causes an immediate illusion of touching a harder object that does not depend on the gradual development of the predictive modulation of grip force.

*For correspondence:
nisky@bgu.ac.il

## Introduction

During everyday interactions with objects, we control and sense the position of these objects and the forces they exert on us. This occurs in perceptual tasks, such as when assessing the stiffness of an object using a tool (*LaMotte, 2000*), and in actions, such as when manipulating the same tool during probing of the object while adjusting the grip force – the perpendicular force between the digits and the object. There is a constant interaction between the perceptual and the motor systems. For example, perception of the mechanical properties of the environment is important for planning future actions. At the same time, we actively explore the environment, obtaining haptic perceptual information by moving and probing our surroundings. Since we do not possess sensors for mechanical impedance, the perception of the impedance of objects, such as their stiffness, damping, and inertia, is based on the integration of motion and force signals, which are sensed during contact with the environment (*Jones and Hunter, 1993*; *Nisky et al., 2008*; *Kuschel et al., 2010*; *Nisky et al., 2010*; *Gurari et al., 2012*).

Our body has two major types of force sensors – kinesthetic and tactile. When holding a pen or a scalpel, kinesthetic position and force information are generated by muscle spindles (length and shortening velocity of the muscles) and Golgi tendon organs (tension in the tendon), respectively. Tactile information arises from several types of cutaneous mechanoreceptors that respond to skin deformation (*Kandel, 2000*). Individuals with impaired tactile sensitivity are deprived of important

information required for planning and controlling object manipulation, such as the magnitude and direction of contact forces, the shape of contact surfaces, and friction (*Johansson and Flanagan, 2009*). Using anesthesia and passive stimulation via robots, the unique role of tactile information in the discrimination of subtle differences in softness has been identified (*Srinivasan and LaMotte, 1995*). However, passive or anesthetized touch is different from active touch with intact tactile sensing (*Gwilliam et al., 2014*), and the combination of kinesthetic and tactile information in perception and action is still poorly understood. In this paper, we investigate how these information modalities are combined in the formation of stiffness perception and control of grip force during intact active touch.

There have been significant advances in the development of mechatronic devices for tactile stimulation of the finger pads (*Provancher and Sylvester, 2009*; *Prattichizzo et al., 2012*; *Quek et al., 2013*; *Quek et al., 2014*). In most of these devices, a tactor (a pin or a platform with a flat high-friction top) moves against the skin; the movement of the tactor deforms the skin. This artificial stretching of the skin emulates the deformation that occurs during interaction with real objects. Previous studies have made use of these devices to investigate the combination of tactile and kinesthetic information in a variety of scenarios. For example, tactor-induced skin-stretch has successfully conveyed direction information (*Gleeson et al., 2009*), navigation cues (*Quek et al., 2015a*) and stiffness information (*Schorr et al., 2015*) in sensory substitution studies. Additionally, adding artificial skin-stretch to kinesthetic force has been shown to augment the perceived friction (*Provancher and Sylvester, 2009*) and perceived stiffness (*Quek et al., 2014*). Meaning, stretching the skin more than it would be during natural interactions, causes the object to feel stiffer than it really is. These studies, however, do not provide any information about the time-course of the formation of the perceptual illusions caused by the skin-stretch, or about the role of skin-stretch in grip force control.

During tool-mediated interactions with objects, humans apply grip force perpendicularly between their fingers and an object to prevent its slippage. When interacting with an object, three known components contribute to our grip force: a predictive feedforward component, which consists of a baseline grip force and a modulation of grip force in anticipation of the load force; and a reactive feedback component that responds to slippage. Feedforward control is used to adjust our grip force in accordance with the expected slipperiness and weight (or load force) of the object (*Flanagan and Wing, 1990*; *Flanagan and Wing, 1997*; *Flanagan et al., 2003*; *Danion and Sarlegna, 2007*; *Danion et al., 2009*; *Leib et al., 2015*; *Leib et al., 2018*; *White et al., 2018*). This adjustment often starts in anticipation to the change in load force (*Forssberg et al., 1992*), and several recent studies revealed that the intended peak grip force can be predicted from the grip force and the rate of its change at the time of initial contact with the object (*White et al., 2011*; *Kuling et al., 2019*). Baseline grip force is maintained to create a safety margin, and is increased when one experiences uncertainty regarding the load force (*Gibo et al., 2013*; *Hadjiosif and Smith, 2015*). Predictive modulation is calculated based on an internal representation of the object dynamics, which is learned by the nervous system (*Johansson and Cole, 1992*; *Shadmehr and Mussa-Ivaldi, 1994*; *Kawato, 1999*; *Davidson and Wolpert, 2004*) and is updated during repeated interactions (*Donchin et al., 2003*). Finally, if cutaneous receptors detect that slippage is occurring, the grip force is increased through rapid feedback control (*Kandel, 2000*; *Johansson and Flanagan, 2009*).

Tactile information can be completely removed by digital anesthesia, and by doing so, has been shown to be necessary for scaling the grip force strength. Additionally, a recent rTMS study suggested that the left supplementary motor area, known to be involved in processing tactile information (*Romo et al., 1993*; *Kim et al., 2015*), is involved in the scaling of grip force (*White et al., 2013*). The timing, on the other hand, can be predictively regulated by kinesthetic information alone (*Nowak et al., 2001*; *Witney et al., 2004*). In the complete absence of force information, grip force is not modulated in accordance with the load force, and the baseline grip force is increased to prevent slippage (*Gibo et al., 2013*).

Skin-stretch devices partially disrupt the natural coupling between tactile and kinesthetic information, and can therefore be used to investigate the contribution of tactile and kinesthetic information to grip force control, while all sensory and motor components are intact. Two recent studies using this technology yielded conflicting results regarding the effects of skin-stretch on grip force, reporting an increased (*Quek et al., 2015a*) and unchanged (*Quek et al., 2015b*) mean grip force, respectively. However, it is unknown how artificial skin-stretch affects the predictive modulation of grip

force in anticipation of load force. Our first goal was therefore to understand the contribution of tactile information to the predictive grip force control.

There are many examples of the dissociation of sensory information processing for perception and action during grasping (*Aglioti et al., 1995*; *Haffenden et al., 2001*) and lifting of objects (*Flanagan and Beltzner, 2000*; *Bringoux et al., 2012*). For example, the size-weight illusion affects the scaling of grip force in the first trials of a lifting task, but with repeated lifting movements the motor effect fades away, while the perceptual illusion persists (*Flanagan and Beltzner, 2000*). During interactions with linear elastic force fields, a delay in the force feedback produces the illusion of touching a softer spring, but the participants' grip force is predictively adjusted to the correct stiffness level and timing (*Leib et al., 2015*; *Leib et al., 2018*). Conversely, introducing a delay in visual feedback led to a persistent alteration of grip force control due to an illusory change in object dynamics (*Sarlegna et al., 2010*). As there is a lack of consistency regarding the question of dissociation between perception and action, it remains to be determined whether the perceptual augmentation of stiffness due to the addition of artificial skin-stretch would affect the predictive grip force adjustment. If so, it remains to be determined whether the perceptual augmentation of stiffness is a direct result of the tactile stimulation, or if it develops with the internal representation that is used for the predictive modulation of grip force in anticipation of load force. Therefore, our second goal in this study was to understand the time-course of developing the perceptual illusion of touching a harder object due to artificial skin-stretch stimulation.

We therefore designed two experiments to test the effect of artificial skin-stretch applied together with kinesthetic force, on the two different components of the predictive control of grip force, and on the formation of stiffness perception. In both experiments, participants were asked to judge the stiffness of virtual objects. These objects were elastic force fields with different levels of stiffness, achieved with a kinesthetic haptic device, and augmented with additional stretch stimuli to the skin of the fingertips. We first focused on how the additional stretch of the fingertips affects the feedforward grip force control. If perception and the predictive control of grip force share similar (or even mutual) stiffness estimation mechanisms, we would expect to find a greater anticipatory modulation of grip force with the load force after repeated interactions with the force field. In addition, the skin-stretch may also increase participants' uncertainty regarding the load force, in which case we would expect to find an increase in the baseline grip force due to an increase in the safety margin. Our second experiment was designed to understand whether our perception of stiffness is influenced by the artificial stretch stimulation immediately, or after the development of the modulation of grip force in anticipation of load force.

## Results

### Experiment 1

### Skin-stretch immediately increased the baseline grip force and gradually increased the feedforward modulation of the grip force with the anticipated load force

In this experiment, we examined the effect of adding artificial skin-stretch to kinesthetic load force on the predictive control of grip force. Participants (N = 10) sat in front of a virtual reality setup and interacted with virtual elastic force fields which were created by a haptic device. In each trial, participants used structured probing movements to evaluate the stiffness of two different force fields, designated *standard* and *comparison*, and decided which had a higher level of stiffness. When interacting with the *standard* force field, in addition to the load force feedback, which had a constant stiffness value of 85 N/m, the skin of the thumb and index finger was stretched using a skin-stretch device (*Figure 1*). The stiffness of the *comparison* force field in each trial was chosen to be one of three different stiffness values (40, 85, and 130 N/m). The magnitude of both the stretch and the load force was proportional to the penetration depth into the virtual elastic force field. The gain of the skin-stretch stimulation (i.e. the amount of skin-stretch relative to the penetration depth) in each trial was chosen from four possible values (0, 33, 66, and 100 mm/m). In each trial, participants made eight discrete movements into each of the two force fields.

In some of the trials, either the second or the seventh probing movements into the *standard* force field, were stretch-catch probes. In these stretch-catch probes, we maintained the load force but

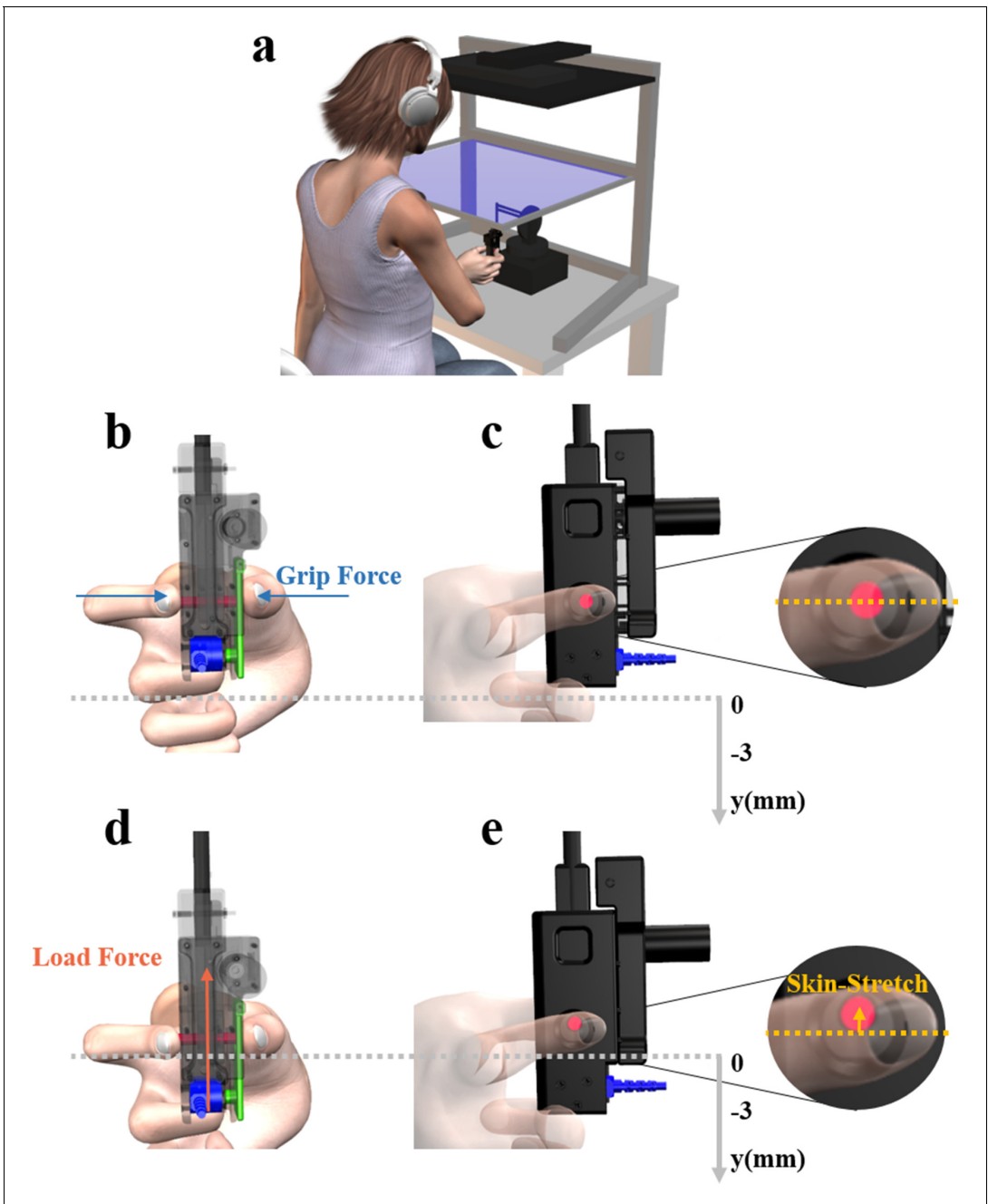

**Figure 1.** Experimental system. (a) The participants sat in front of a virtual reality rig, and held the skin-stretch device, which was mounted on the end of a haptic device. (b) Back and (c) Side views of the skin-stretch device in the case of no interaction with the force field, and therefore the load force and the skin-stretch were zero. Zoomed in view of (c) illustrates that in this case the tactor (red rod) was in its zero state. (d) Back and (e) Side views of the skin-stretch device during interaction with the force field. Both the load force and the skin-stretch increased with the penetration into the force field. Zoomed in view of (d) shows the upward movement of the tactor in this case. A force sensor (blue) was embedded in the device to measure the grip force that the participants applied, via the lever (green) which transmitted the grip force from the contact point to the sensor.

omitted the skin-stretch. The purpose of the stretch-catch probes was to allow us to investigate the predictive component of grip force control. To identify the predictive component, which is expected to develop with repeated stretch stimulation, we compared between the grip force that was applied in the second and the seventh stretch-catch probes. Additionally, we used the applied grip force in the stretch-catch probes to fit a model that predicts the *intended peak grip force* (as in *White et al.,*

*2011*; *Kuling et al., 2019*). We used this model to assess the evolution of the predictive grip force over repeated probing movements.

## Analyses of stretch-catch probes

The following analyses were done only on the stretch-catch probes which appear in some of the second and seventh probing movements. *Figure 2* presents examples of grip force, load force, and tactor displacement trajectories from both regular probes, and stretch-catch probes. These trajectories clearly show that despite the skin-stretch stimulation, participants maintained the grip force-load force modulation. In *Figure 2c and d*, we can see that during probes with high levels of tactor displacement gain, the grip force signals have a double peak pattern. We quantified these reactive responses, and thoroughly investigated them with additional control experiments (see Appendix 1). However, our conclusions about these patterns are speculative and may be specific to our device, and therefore, we chose to defer them to Appendix 1, and focused our current analyses and conclusions on the predictive response alone.

*Figure 2c and d* also show that in the event of large tactor displacement gains (66 and 100 mm/m), repeated interactions with the force field led to an increase in the predictive grip force control. The grip force trajectories in the stretch-catch probes in panels (**b**)– (**d**) show that the feedforward predictive component of the grip force (in anticipation of the skin-stretch stimulus that does not occur) is characterized by a single peak. Moreover, in the seventh probing movement, this peak grip force is larger than it would be if no skin-stretch was expected [compare the seventh movements of panel (**d**) and (**a**)].

To quantify these effects, we analyzed the *peak grip force-load force ratio* of the predictive grip force (*Figure 3*). In trials without skin-stretch, that is, with a gain of 0 mm/m (*Figure 3a and b*), the *peak grip force-peak load force ratio* decreased between the first and the seventh probing movements (rm-regression, main effect of 'probing movement': $F_{(1,9)} = 11.75, \quad p = 0.0075$). This observed decrease is consistent with previous results reported in the literature (*Leib et al., 2015*; *Leib et al., 2018*). A likely cause is that repeated interactions with the elastic force field may have improved participants' certainty about the internal representation of the force field, allowing them to lower their grip force.

Our next step was to isolate the effect of the skin-stretch on the predictive component of the grip force. To do so, we analyzed the grip force-load force trajectories of the stretch-catch probes. *Figure 3c* presents the average grip force trajectories divided by the peak load force as a function of time in the second and seventh probing movements for different tactor displacement gains. We saw that the amplitude of the average grip force trajectories decreased between the second and seventh probing movements. When comparing between the trajectories of probes with different tactor displacement gains, we observed an increase with the gain value. This increase was evident in both the second and seventh probing movements but was greater for the seventh probing movement. To quantify these qualitative observations, we calculated the *peak grip force-peak load force ratio* for the stretch-catch probes for each of the tactor displacement gains (*Figure 3d*). We observed a rising trend in the applied grip force with higher values of the expected tactor displacement gain, for both the second (gentle rise) and seventh (steep rise) probing movements. That is, participants applied more grip force per amount of kinesthetic load force when interacting with force fields with higher levels of tactor displacement gain. This trend is significantly higher in the seventh probe compared to the second (rm-General Linear Model, main effect of 'gain': $F_{(1,9)} = 5.67, \quad p = 0.0411$; main effect of 'probing movement': $F_{(1,9)} = 8.50, \quad p = 0.0172$; interaction between 'gain' and 'probing movement' variables: $F_{(1,9)} = 6.92, \quad p = 0.0273$). The gentle increase in the second probing movement can be explained by the fact that participants had already been exposed to the skin-stretch stimulus in the first movement, and could therefore predict the stimulus. The steep increase in the seventh probing movement suggests that with repeated interactions, the participants continued to build up the increase in the anticipatory control of their applied grip force.

While the *peak grip force-peak load force ratio* analysis revealed the effect of the skin-stretch on the predictive component of the grip force, it did not differentiate between the effect on each of the two components of the predictive grip force, the baseline and the modulation. The **baseline** provides a safety margin against slippage and depends on the certainty regarding the force field. The **modulation** is adjusted in anticipation of the load force, and depends on one's estimation of the

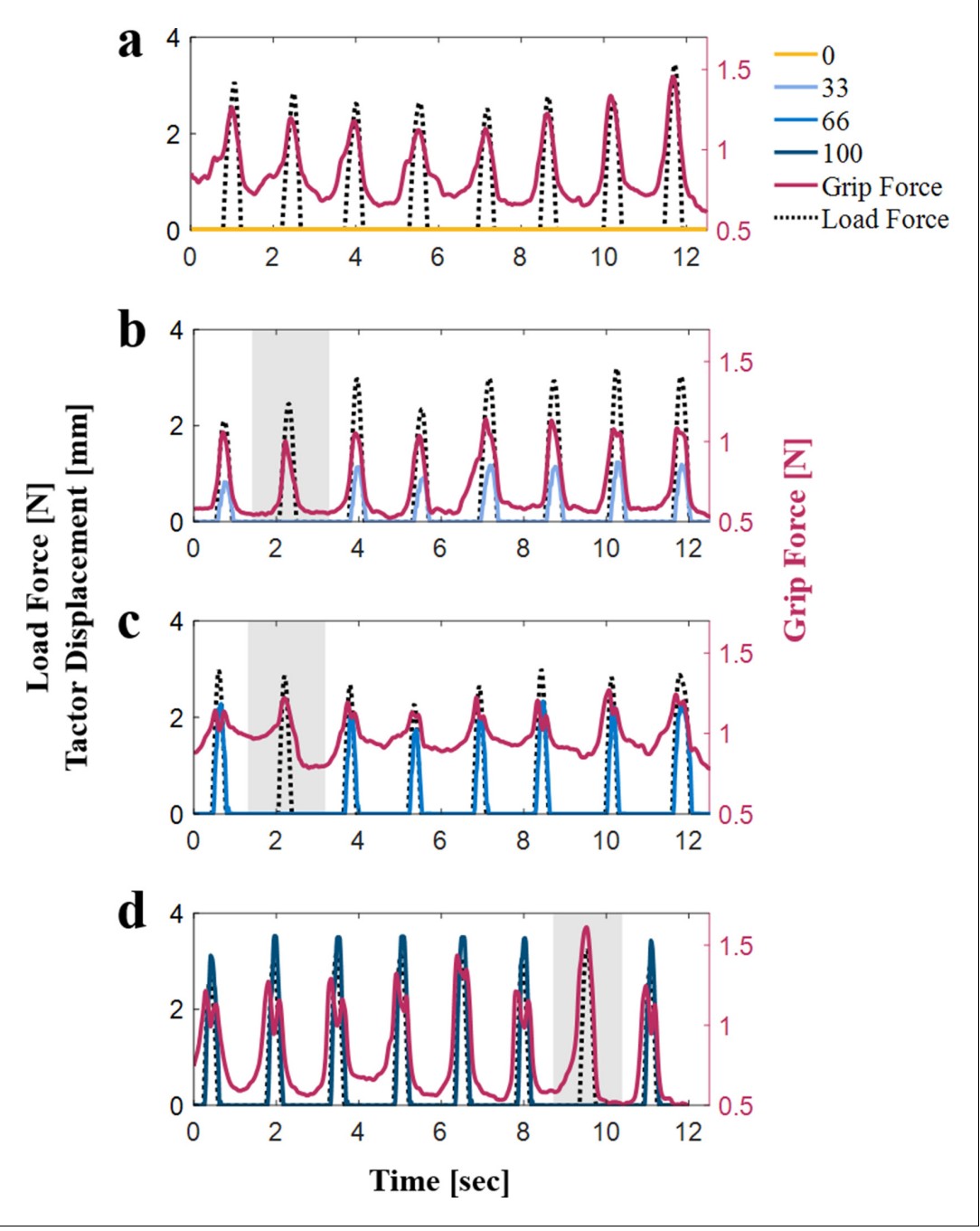

**Figure 2.** Examples of load force (black dotted line), grip force (red solid line), and tactor displacement (different shades of blue) trajectories of a typical participant. The gray shaded regions highlight stretch-catch probes, in which we maintained the load force but surprisingly omitted the skin-stretch. (a) Tactor displacement gain of 0 mm/m. (b) Tactor displacement gain of 33 mm/m. (c) Tactor displacement gain of 66 mm/m. (d) Tactor displacement gain of 100 mm/m.

mechanical impedance of the force field and slipperiness of the finger interface (*Flanagan and Wing, 1990*; *Leib et al., 2015*; *Leib et al., 2018*). To separate between the effect of the skin-stretch on each of these two components, we performed a linear regression analysis in the grip force-load force plane (*Figure 4a and d*), similar to the analysis in *Flanagan and Wing (1990)*; *Leib et al. (2015)*; *Leib et al. (2018)*. The intercept of this regression is close to the baseline grip force; i.e. the amount of grip force that was applied by the participants when no external load force was applied

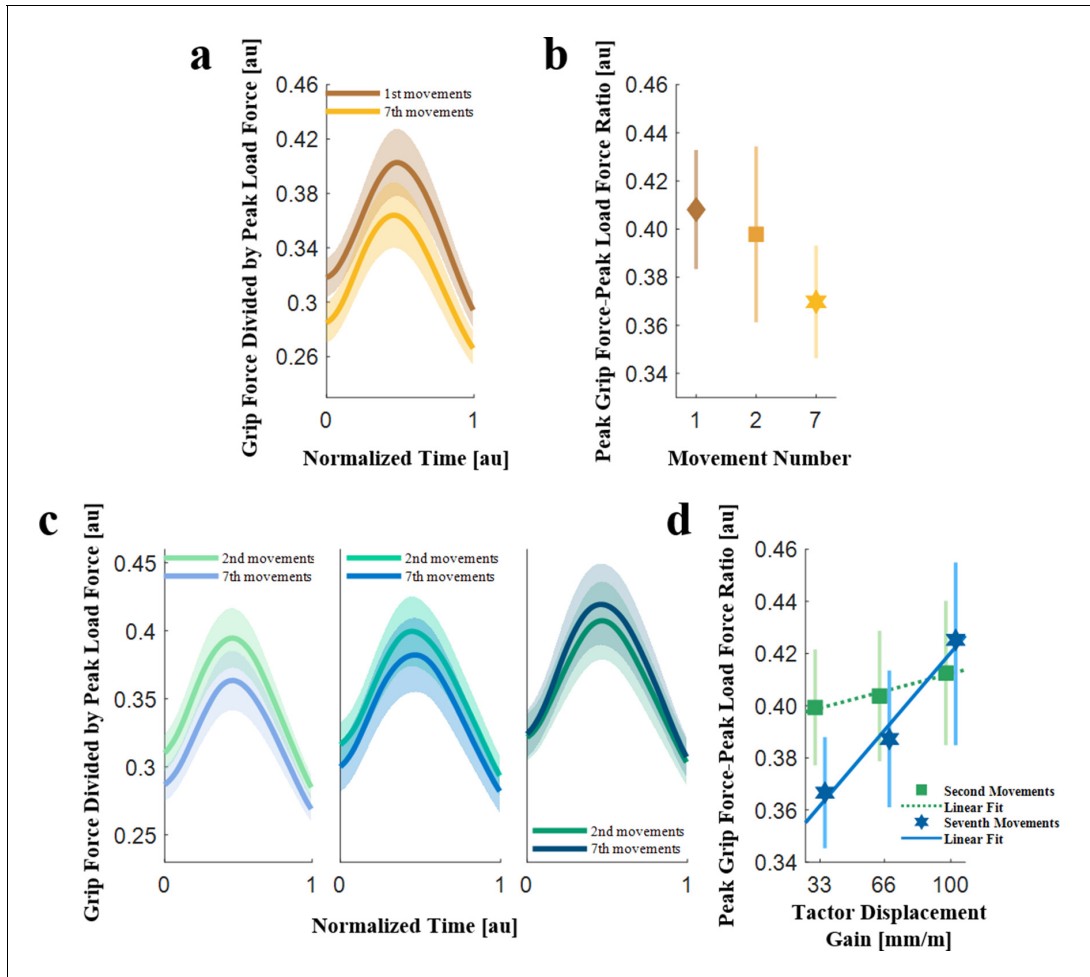

**Figure 3.** Grip force-load force ratios (N = 10). (a) The grip force trajectories divided by the peak load force averaged across all the participants, for the first (brown) and seventh (orange) probing movements in the trials with no skin-stretch. The trajectories were time-normalized and aligned such that 0 was the onset of the contact with the force field, and one was the end of the interaction. The shading represents the standard errors. (b) The peak grip force-peak load force ratio in the first, second, and seventh probing movements. The markers and the vertical lines represent the peak grip force-peak load force ratios averaged across all the participants, and their standard errors, respectively. (c) The grip force trajectories divided by the peak load force averaged across all the participants, for the second (green) and seventh (blue) probing movements in trials with skin-stretch. (left: tactor displacement gain of 33 mm/m, middle: 66 mm/m, and right: 100 mm/m). The trajectories were time-normalized and aligned such that 0 was the onset of the contact with the force field, and one was the end of the interaction. The shading represents the standard errors. (d) The peak grip force-peak load force ratio in the second (green) and seventh (blue) probing movements in the trials with skin-stretch, as a function of the tactor displacement gain. The markers and the vertical lines represent the peak grip force-peak load force ratios averaged across all the participants, and their standard errors, respectively. The dotted green and solid blue lines represent the average fitted regression lines.

by the haptic device. The slope of the regression represents the modulation of the grip force in anticipation of the load force, that is, the amount of change in the grip force per change in the load force during the interaction with the force field. Prior works that studied interaction with elastic objects linked the slope to both the slip ratio and the anticipated load force (*Flanagan and Wing, 1990*; *Leib et al., 2015*; *Leib et al., 2018*).

In trials without skin-stretch; that is, with a gain of 0 mm/m, the intercept decreased with repeated probing movements (rm-regression, main effect of 'probing movement': $F_{(2,18)} = 10.98$, $p = 0.0008$), while the slope did not change (rm-regression, main effect of 'probing movement': $F_{(2,18)} = 2.63$, $p = 0.0994$) (*Figure 4a, b and c*). As the intercept represents the applied safety margin, this decrease indicates that participants became more confident about their estimation of the stiffness of the elastic force fields and the slipperiness of the contact. The lack of change in the slope suggests that there was little or no change in the represented stiffness or slip ratio.

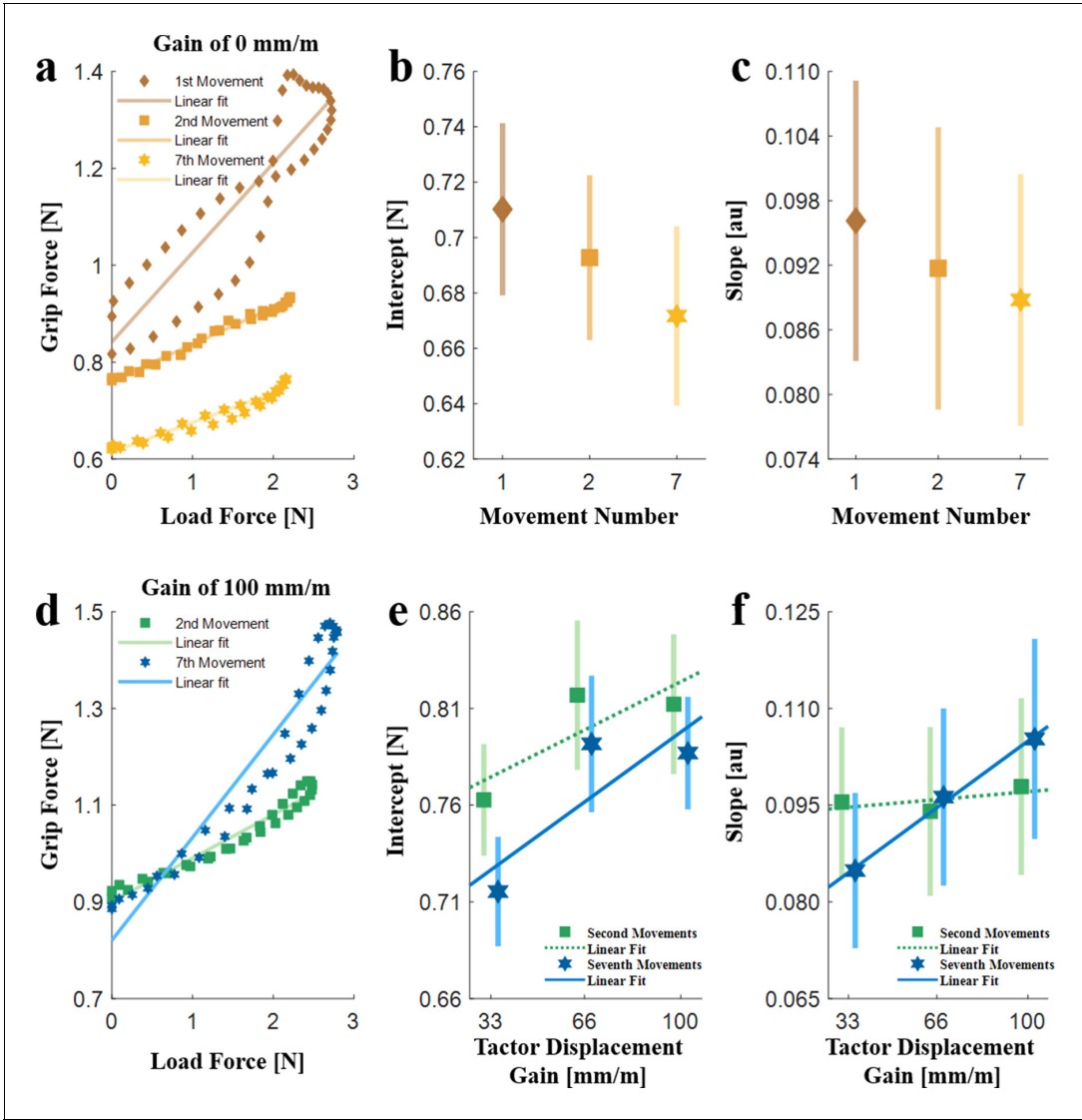

**Figure 4.** The linear regression analysis (N = 10). (a) An example of grip force-load force regression of the first, second and seventh probing movements in a trial with a gain of 0 mm/m. (b) Intercept and (c) Slope of the regression for the 0 mm/m gain trials averaged across all the participants in the first, second, and seventh probing movements. (d) An example of grip force-load force regression of the second and seventh probing movements in a trial with a gain of 100 mm/m. (e) Intercept and (f) Slope, for the different levels of the tactor displacement gain. The dotted green and solid blue lines represent the average fitted regression lines. The markers and the vertical lines represent the values averaged across all the participants, and their standard errors, respectively.

*Figure 4d* presents an example of a *grip force-load force regression* analysis of the second and seventh probing movements of a typical participant in a trial with a tactor displacement gain of 100 mm/m. *Figure 4e and f* present the *grip force-load force regression* analysis (intercept and slope) as a function of the tactor displacement gain. *Figure 4e* shows that the intercept increased with an increase in the expected tactor displacement gain (rm-General Linear Model, main effect of 'gain': $F_{(1,9)} = 7.95, \quad p = 0.0201$). This suggests that the addition of the artificial skin-stretch increased the participants' uncertainty about their internal representation of the elastic force fields. Surprisingly, we did not find a difference between the second and seventh probing movements (rm-General Linear Model, main effect of 'probing movement': $F_{(1,9)} = 2.46, \quad p = 0.1514$; interaction between 'gain' and 'probing movement' variables: $F_{(1,9)} = 0.44, \quad p = 0.5224$).

*Figure 4f* shows that the slope of the second probing movement did not depend on the expected tactor displacement gain, while the slope of the seventh probing movement increased

with an increase in the expected tactor displacement gain (rm-General Linear Model, main effect of 'gain': $F_{(1,9)} = 5.17$, $p = 0.0491$; main effect of 'probing movement': $F_{(1,9)} = 8.20$, $p = 0.0187$; interaction between 'gain' and 'probing movement' variables: $F_{(1,9)} = 5.33$, $p = 0.0463$). This suggests that following repeated exposure to the artificial skin-stretch stimulation, the participants either increased their estimation of the expected force (likely due to the increase in the representation of the stiffness of the force field) or of the slipperiness of the device. It is important to mention that the slope values are small with respect to previous studies (*Flanagan and Wing, 1990*; *Leib et al., 2015*) as our device only measures a downscaled version of the grip force (see methods and Appendix 2 for more information).

## The evolution of the predictive grip force over repeated probing movements

While providing a clear picture about the predictive control of grip force in the stretch-catch probes, the linear regression analysis does not reveal the evolution of the different grip force components with repeated probing movements into the force field within each trial. Therefore, we used the approach in *White et al. (2011)*; *Kuling et al. (2019)* to analyze all the probing movements (*Figure 5*), rather than only the second and seventh probes which can be analyzed via the stretch-catch probes. We calculated the *grip force baseline* between two consecutive movements, and used the stretch-catch probes to calculate the coefficients of a multiple regression model that predicts the *intended peak grip force* based on the *grip force at contact* (that is, the initial contact with the elastic force field) and its rate of change. It is important to note that the *intended peak grip force* is affected by both the baseline and a modulation of the grip force in anticipation of the load force. Therefore, to better assess the evolution of these two very different components of predictive grip force control, and to compare the results of this analysis to those of the *grip force-load force regression* analysis that we performed on the stretch-catch probes (*Figure 4*), we further separate this analysis into *grip force at contact* and *grip force modulation*. We calculated the *grip force modulation* by subtracting the *grip force at contact* from the *intended peak grip force*. Finally, to isolate the effect of the artificial stretch stimulation on each of these grip force components (illustrated in *Figure 5*), we calculated the difference between the values obtained due to the artificial stretch and those of the 0 mm/m tactor displacement gain. Meaning, for each participant, probing movement, and gain, we subtracted the respective grip force value at the zero gain, and present this difference in *Figure 6*.

In *Figure 5a*, the *grip force baseline* is depicted as a function of the movement number within a trial, for the four different levels of tactor displacement gain (0, 33, 66, and 100 mm/m). In the first probing movement, the *baseline* grip force values of all the tactor displacement gains are similar. However, already in the second movement a difference between the *grip force baseline* values of the different tactor displacement gains can be observed. The size of this difference varied between the different gains (*Figure 6a*), but for each gain, it was maintained roughly constant throughout the remaining probing movements (rm-General Linear Model, main effect of 'gain': $F_{(1, 9)} = 17.96$, $p = 0.0022$; main effect of 'probing movement': $F_{(6, 54)} = 3.84$, $p = 0.0029$; interaction between 'gain' and 'probing movement' variables: $F_{(6, 54)} = 7.09$, $p<0.0001$). Meaning, due to the additional tactile stimulation, the *grip force baseline* increased in accordance to the amount of stretch immediately following the first probing movement. These results are similar to those of the intercept of the *grip force-load force regression* analysis (*Figure 4*); a difference between the intercept of the different tactor displacement gains is observed even from the second probing movement, whereas no significant difference between the intercept of the second and seventh movements was found. The significant effect of 'probing movement' seen here was not observed in the analysis of the *grip force-load force regression* intercept since the major difference is between the first and second probing movements, and thus, could not be exposed using an analysis of the stretch-catch probes in the second and seventh probing movements.

*Figure 5b* shows the *intended peak grip force* as a function of the movement number within a trial, for the four different tactor displacement gains. Here too, the *intended peak grip force* was similar between the different tactor displacement gains in the first probing movement. From the second probing movement onward, a difference between the *intended peak grip force* of the different tactor displacement gains can be seen, which increased slightly with repeated interactions

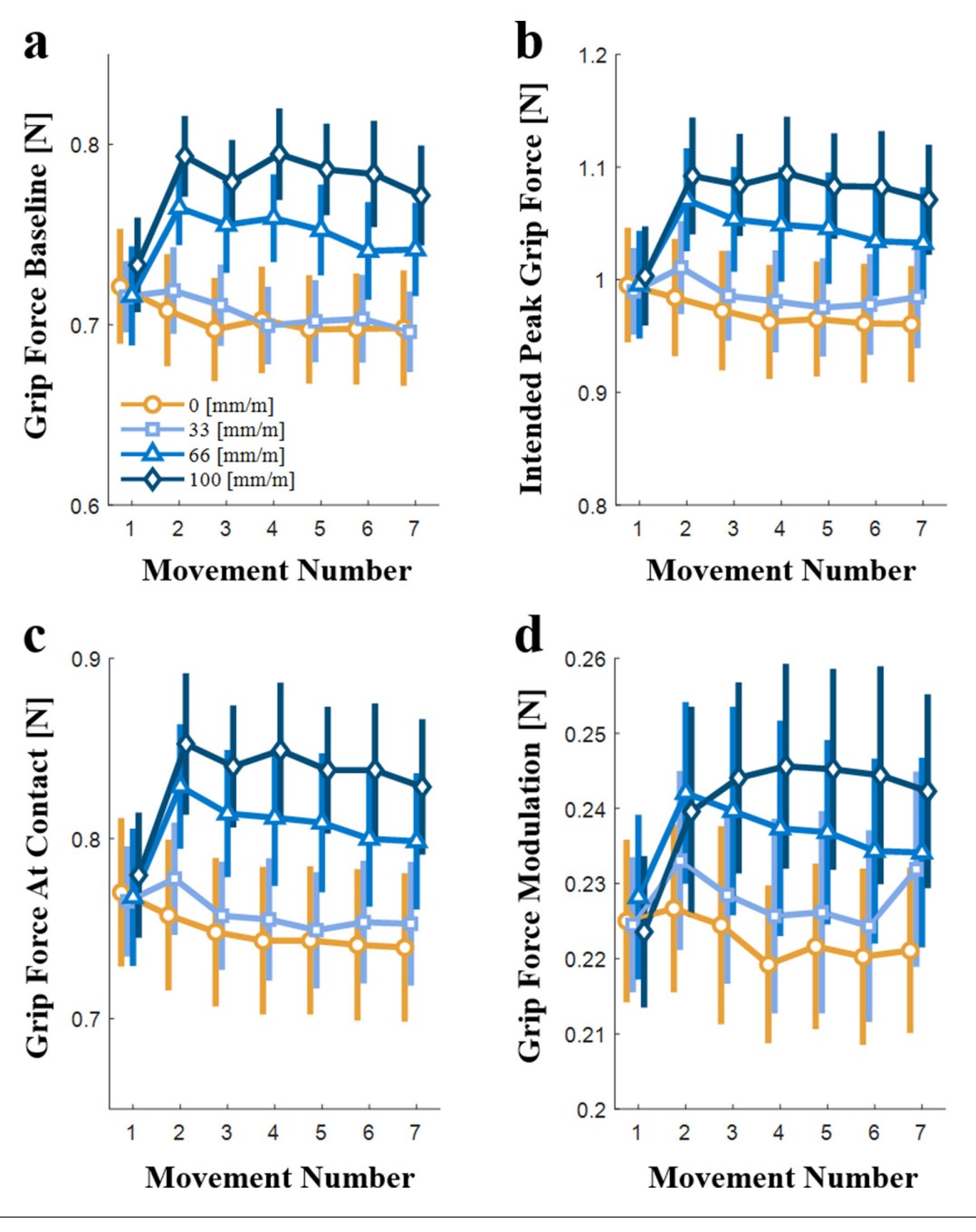

**Figure 5.** Evolution of the predictive control of grip force with repeated interaction (N=10). (a) The grip force baseline, (b) the intended peak grip force, (c) the grip force at contact, and (d) the grip force modulation during the interaction with the elastic force field, all as a function of the movement number. The yellow lines and symbols represent trials with tactor displacement gain of 0 mm/m, and the different shades of blue represent positive tactor displacement gains (33, 66, and 100 mm/m). Each symbol is the average value for all the participants, and the vertical lines represent the standard errors.

(*Figure 6b*), especially in the largest gain (rm-General Linear Model, main effect of 'gain': $F_{(1,9)} = 14.31$, $p = 0.0043$; main effect of 'probing movement': $F_{(6, 54)} = 7.96$, $p<0.0001$; interaction between 'gain' and 'probing movement' variables: $F_{(6, 54)} = 7.42$, $p<0.0001$). This result is similar to that of the *peak grip force-peak load force ratio* (*Figure 3d*) in which we saw a moderate increase of the ratio in the second movement and a deep increase in the seventh movement.

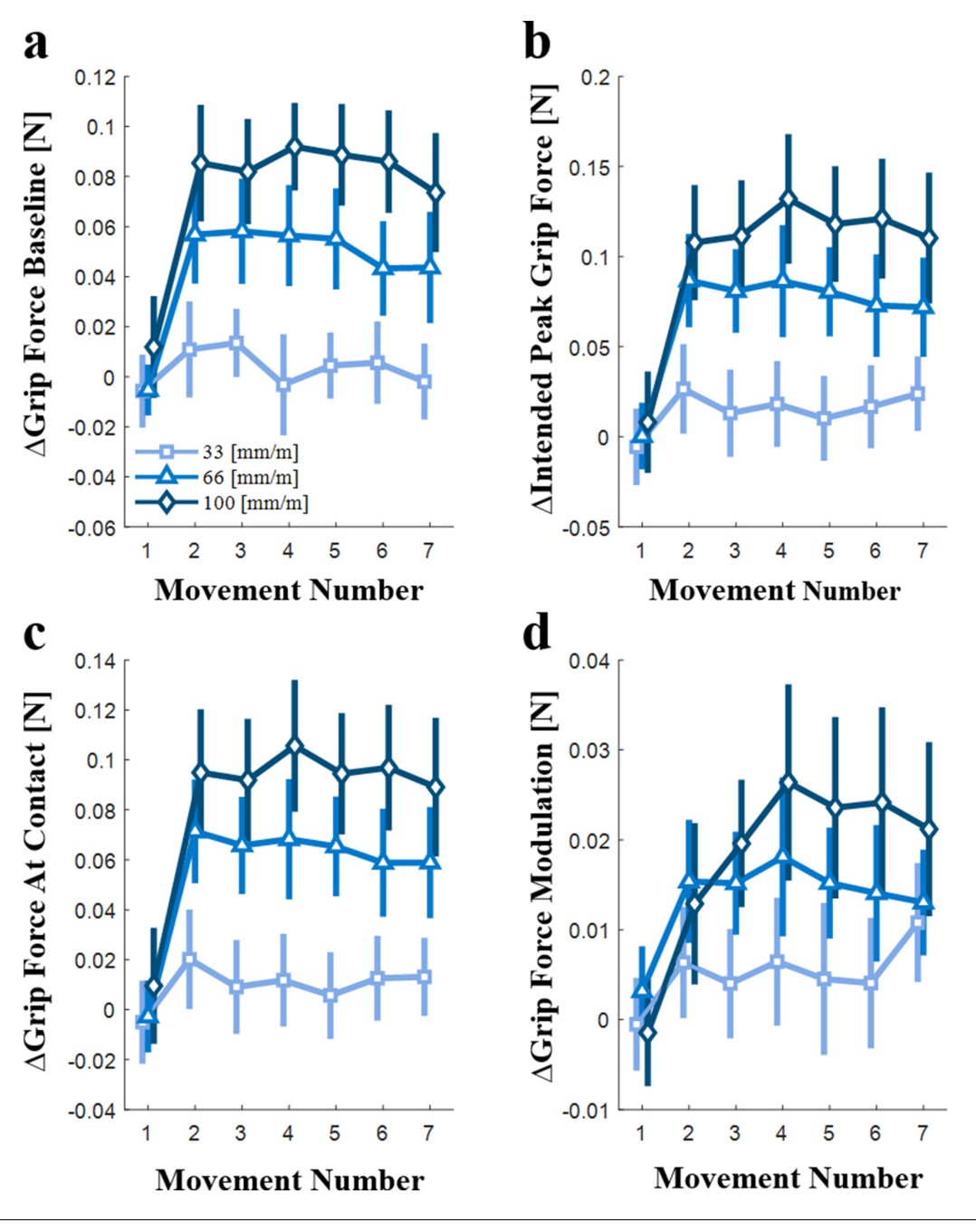

**Figure 6.** The effect of artificial stretch stimulation on the evolution of the predictive control of grip force with repeated interaction (N=10). The difference between the effect of positive tactor displacements gains (33, 66, and 100 mm/m) and the 0 mm/m tactor displacement gain on (**a**) the grip force baseline, (**b**) the intended peak grip force, (**c**) the grip force at contact, and (**d**) the grip force modulation during interaction with the elastic force field, all as a function of the movement number. The different shades of blue represent the different positive tactor displacement gains (33, 66, and 100 mm/m). Each symbol is the average value for all the participants (N = 10), and the vertical lines represent the standard error.

We next separated the *intended peak grip force* into the *grip force at contact* (*Figures 5c* and *6c*) and *grip force modulation* (*Figures 5d* and *6d*) components. The *grip force at contact* exhibited a similar trend to that of both the *grip force baseline* (*Figure 5a*) and the intercept of the *grip force-load force regression* analysis (*Figure 4e*) (rm-General Linear Model, main effect of 'gain':

$F_{(1,9)} = 15.97$, $p = 0.0031$; main effect of 'probing movement': $F_{(6, 54)} = 7.51$, $p<0.0001$; interaction between 'gain' and 'probing movement' variables: $F_{(6, 54)} = 7.65$, $p<0.0001$). In contrast, the *grip force modulation* reveals a different trend. The first probing movement is similar for all four tactor displacement gains, and it is followed by a gradual change with repeated probing movements. This gradual development is different for each gain level (***Figure 6d***) and is especially clear for the larger 100 mm/m gain. (rm-General Linear Model, main effect of 'gain': $F_{(1,9)} = 6.22$, $p = 0.0342$; main effect of 'probing movement': $F_{(6, 54)} = 5.69$ $p = 0.0001$; interaction between 'gain' and 'probing movement' variables: $F_{(6, 54)} = 4.48$, $p = 0.0010$). This result corroborates our *grip force-load force regression* slope analysis (***Figure 4f***), in which in the second probing movement the dependence of the slope on the tactor displacement gain was minor, whereas in the seventh probing movement there was a significant dependence of the slope on the tactor displacement gain. This analysis enables us to pinpoint the time-course of the change with greater precision than the stretch-catch probe analysis; calculating the *grip force modulation* showed that the gradual change culminates after four probing movements.

To summarize, we found that following exposure to high levels of artificial skin-stretch participants immediately increased their safety margin and maintained it even after many interactions. They also increased the predictive grip force modulation in anticipation of the load force. This increase developed gradually with subsequent probing interactions with the same elastic force field.

## Experiment 2

### The perceptual illusion of touching a harder object is immediate, and does not depend on the development of the predictive control of grip force

Previous studies (***Quek et al., 2013***; ***Quek et al., 2014***) showed that when participants freely explored (used as many probing movements as they wished) the stiffness of elastic force fields, adding artificial skin-stretch caused an increase in the perceived stiffness. Moreover, they found this perceptual augmentation to be linearly related to the tactor displacement gain (that is, the amount of skin-stretch per movement). In Experiment one, we showed that the addition of the skin-stretch led to an increase in the magnitude of the predictive grip force, which developed over different time scales with repeated interactions: the grip force baseline increased immediately after one probing movement whereas the modulation of the grip force in anticipation of the load force developed gradually over approximately four probing movements. Similar to the perceptual augmentation, the increase in the grip force was also linearly dependent on the expected tactor displacement gain. These two findings presented evidence of artificial skin-stretch leading to both a perceptual augmentation and an increase in the predictive grip force. It remained unclear, however, whether the increase in the perceived stiffness was induced by the tactile stimulus as a high-level cognitive illusion, or a low-level effect that developed following the build-up of the increased feedforward grip force due to repeated interactions with the tactile stimulation. If the first were to be the case, we would expect the perceptual illusion to be created immediately following exposure to the skin-stretch stimulus. In the event of the latter, on the other hand, the perceptual illusion would develop with subsequent probing movements, similar to the increase in the modulation of grip force. To investigate this matter, we examined the timescale for the creation of the perceptual augmentation. We did this by using a stiffness discrimination task, and examining the dependence of the effect of adding artificial skin-stretch to kinesthetic load force on the number of serial interactions with the elastic force field. Participants (N = 20) probed pairs of elastic force fields and determined which one had a higher level of stiffness. The *standard* force field had a constant stiffness value (85 N/m), and was augmented by one of four different tactor displacement gains (0, 33, 66, and 100 mm/m). The stiffness of the *comparison* force field was chosen from a range of ten values, evenly spaced between 40–130 N/m. The participants were divided into two groups: Group 1 (N = 10) could probe each force field only once, and Group 2 (N = 10) used free exploration, meaning they could probe each force field as many times as they desired.

### Skin-stretch caused perceptual overestimation of stiffness

Adding artificial skin-stretch to kinesthetic load force caused participants to overestimate the stiffness of the *standard* force field. We found this by fitting psychometric functions to the probability that the participants would respond that the *comparison* force field had a higher level of stiffness as

a function of the difference between the stiffness levels of the *comparison* and *standard* force fields. Our results show that the augmentation in the perceived stiffness increased linearly as a function of the tactor displacement gain in both groups. The psychometric curves of typical participants from Group 1 and Group 2 (*Figure 7a and d*, respectively) show that artificial skin-stretch caused the illusion of touching a harder object, that is, overestimation of the perceived stiffness. As shown in the psychometric curves, in trials lacking artificial skin-stretch (tactor displacement gain of 0 mm/m), the Point of Subjective Equality (PSE, a measure of bias in the perceived stiffness) was close to zero, and the slope of the psychometric curve was steep, indicating that the participant could accurately distinguish between the stiffness levels of the two force fields. Stretching the finger-pad skin of these participants during their interactions with the *standard* force field led to a rightward shift of the psychometric curves representing these trials, and a positive PSE, indicating that these participants overestimated the stiffness of the *standard* force field. A larger increase in the tactor displacement gain caused an additional rightward shift of the PSE. These shifts of the PSE values occurred without practically any change in the slopes of the psychometric curve, indicating that stretching the finger-pad skin did not affect the stiffness discrimination ability.

## Group 1

An examination of the results of all 10 participants revealed a similar trend to that observed in the psychometric curves of the typical participant; the addition of the artificial skin-stretch led to an increase in the perceived stiffness (as quantified by the PSE) for 8 out of the 10 participants (*Figure 7b*), which was statistically significant (rm-regression, main effect of 'gain': $F_{(1,9)} = 15.38$, $p = 0.0035$). The increase in the average perceived stiffness for a gain of 100 mm/m was 30.73 N/m with a standard error of 2.26 N/m (35% of the kinesthetic stiffness level). In contrast to the observed relation between the PSE and the tactor displacement gain, we did not find any difference in discrimination sensitivity (as quantified by the JND) with an increase in the tactor displacement gain (*Figure 7c*, rm-regression, main effect of 'gain': $F_{(1,9)} = 0.06$, $p = 0.8102$).

## Group 2

The addition of the artificial skin-stretch led to an increase in the perceived stiffness for 9 out of the 10 participants (*Figure 7e*), and was statistically significant (rm-regression, main effect of 'gain': $F_{(1,8)} = 42.47$, $p = 0.0002$). The increase in the average perceived stiffness for a gain of 100 mm/m was 39.10 N/m with a standard error of 1.87 N/m (47% of the kinesthetic stiffness level). The only participant who showed the opposite trend in perceived stiffness (that is, a decrease in the perceived stiffness due to the skin-stretch) reported that she was aware of the applied skin-stretch, and tried to resist its effect by responding the opposite to what she felt. Therefore, this participant was excluded from the figures and the statistical analyses (which did not affect the statistical significance or any other conclusion). Similar to Group 1, this group also exhibited no difference in the discrimination sensitivity with an increase in the tactor displacement gain (*Figure 7f*, rm-regression, main effect of 'gain': $F_{(1,8)} = 2.060$, $p = 0.1890$). Based on this finding from the results of both groups, we therefore conclude that the added tactile stimulation did not impair the discrimination accuracy.

Our goal in this experiment was to assess the timescale for the creation of the perceptual augmentation caused by the artificial skin-stretch. Is the increase in the perceived stiffness caused immediately after experiencing the tactile stimulation, or does it depend on the development of the increased predictive modulation of grip force that builds up with repeated interaction with the exposure to the skin-stretch? A direct comparison between the two groups revealed no differences in the bias in stiffness perception (as quantified by the PSE) or in the discrimination sensitivity (as quantified by the JND) of the two groups. The PSE mean effect of the first group was slightly smaller than that of the second group, but the difference between these effects was not statistically significant (rm-General Linear Model, PSE: main effect of 'group number': $F_{(1,17)} = 1.36$, $p = 0.2595$, JND: main effect of 'group number': $F_{(1,17)} = 0.78$, $p = 0.3904$). We can therefore conclude that the perceptual illusion was formed immediately, after a single probing movement into the force field, and before the increase in the predictive modulation of grip force in anticipation of load force that we identified in Experiment 1 was formed.

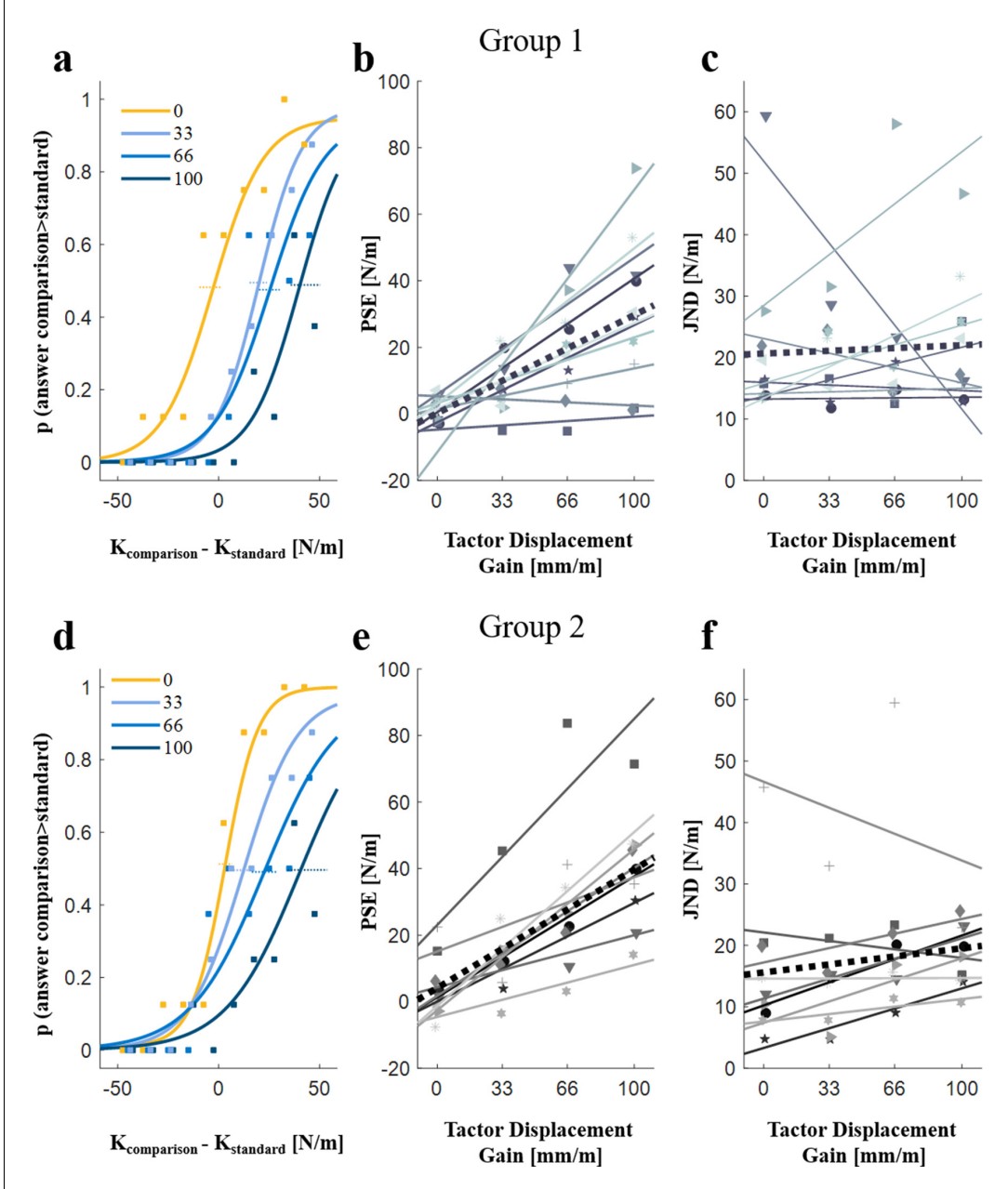

**Figure 7.** The effect of artificial skin-stretch on perception. Example of psychometric curves of typical participants for different levels of tactor displacement gain from (a) Groups 1 and (d) Group 2. The abscissa is the difference between the stiffness levels of the comparison and the standard force fields. The ordinate is the probability of responding that the comparison force field had higher level of stiffness. The four different curves represent the participant's responses for the four different values of the tactor displacement gain (0, 33, 66, and 100 mm/m). The horizontal dotted lines represent the standard errors for the PSE (Point of Subjective Equality).The PSE values as a function of the tactor displacement gain for (b) Group 1 (N = 10) and (e) Group 2 (N = 9).The JND (Just Noticeable Difference) values as a function of the tactor displacement gain for (c) Group 1 (N = 10) and (f) Group 2 (N = 9), respectively. The markers represent the PSE and JND values of each participant. The solid lines show the linear regression of the PSE (b) and (e) or JND (c) and (f) values as a function of the tactor displacement gain for each of the participants, and the black dotted line is the average regression line across all participants.

## Skin-stretch caused an increase in the applied grip force

The main goal of this experiment was to determine the timescale for the creation of the perceptual illusion. However, we also recorded the grip force applied throughout the probing interactions, and investigated the possibility of reproducing some of our findings in Experiment 1 in the more natural

probing setting of Experiment 2, concurrently with pinpointing the number of probing interactions required to form the illusion. *Figure 8* presents examples of grip force, load force, and tactor displacement trajectories of a participant from the free exploration group (Group 2). As in Experiment 1, a visual examination of the trajectories revealed a non-uniform peak pattern that appeared predominantly in trials in which skin-stretch was applied. For example, the grip force signals in

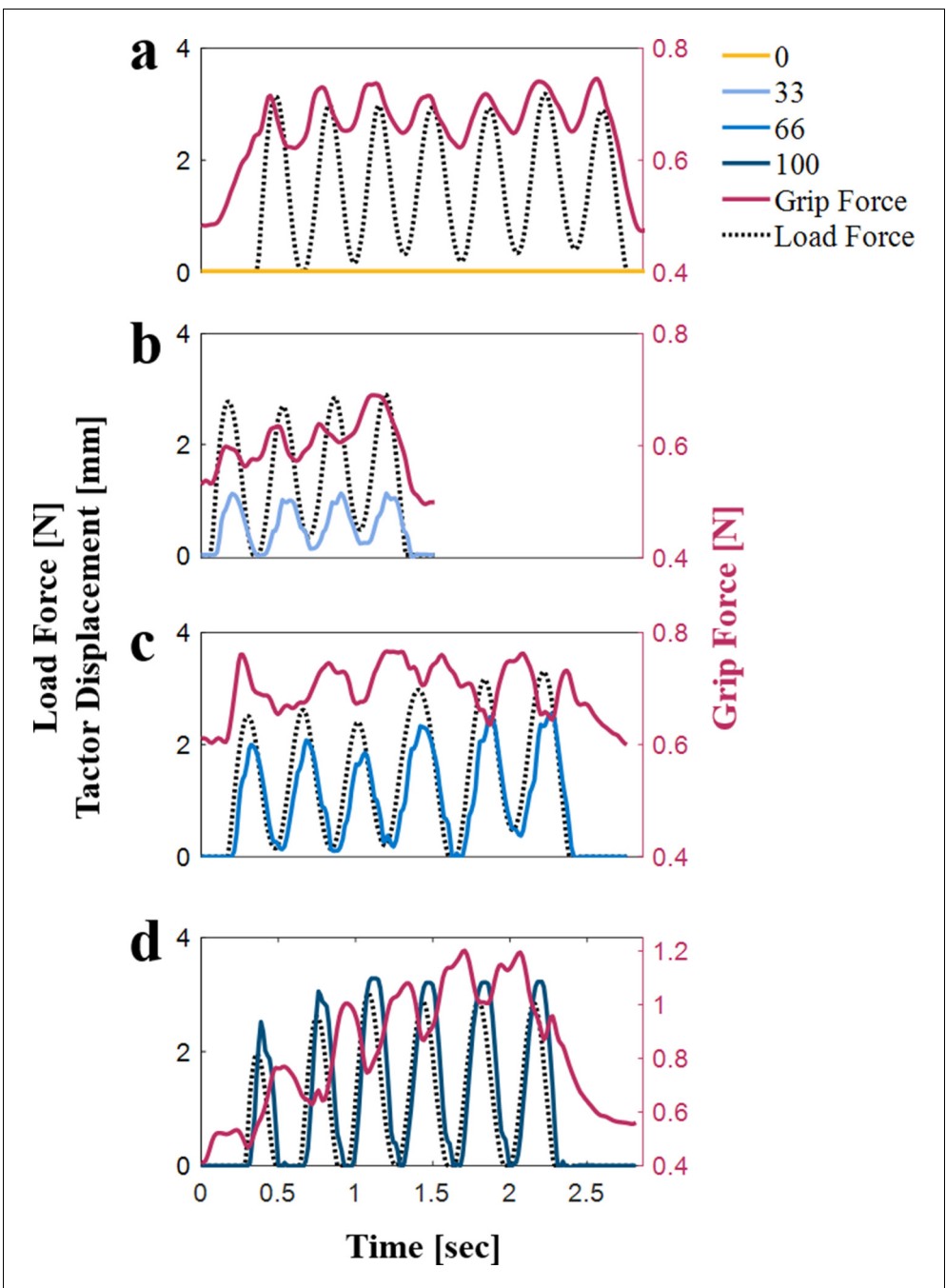

**Figure 8.** Examples of load force (black dotted line), grip force (red solid line), and tactor displacement (different shades of blue) trajectories of a typical participant from the free exploration group (Group 2). (a) Tactor displacement gain of 0 mm/m. (b) Tactor displacement gain of 33 mm/m. (c) Tactor displacement gain of 66 mm/m. (d) Tactor displacement gain of 100 mm/m. The scale of the lower panel is different from the scale of the other panels because we wanted to emphasize the irregular peak pattern that we observed for high levels of tactor displacement gain.

*Figure 8c and d* show that increasing the tactor displacement gain caused several grip force peaks around the load force peaks. In addition, the grip force trajectories give the impression of the presence of phase shifts between grip force responses and the load force due to the added tactor displacement. However, we suggest that in reality participants' grip force in Experiments 1 and 2 were similar. In Experiment 1, participants made discrete probing movements into the force field, leading to the clear grip force trajectories seen in *Figure 2*. In Experiment 2, however, participants used free exploration and the probing movements lacked the separation that existed in Experiment 1. In Experiment 1, we observed two peaks in the grip force trajectories when high tactor displacement gains were applied. We hypothesized that this pattern was due to the reactive grip force component and an artifact specific to our skin-stretch device (see Appendix 1 for more information). We therefore posit that the free exploration used in Experiment 2 caused the second peak of each probe to fuse with the first peak of the subsequent probe. Importantly, *Figure 8* also shows that when high tactor displacement gains were applied, there was a rising trend in the magnitude of the applied grip force as the participant made subsequent probing movements. This phenomenon is illustrated by comparing the 0 and 100 mm/m gain plots; unlike the 0 mm/m plot (*Figure 8a*), a rise in the grip force magnitude can be seen with subsequent probing movements in the 100 mm/m plot (*Figure 8d*).

To quantify the change in the grip force with repeated probing interactions within each trial, we compared between the *peak grip force-peak load force ratio* in the first and last probing movements. Unlike Experiment 1, in which we separated the predictive component into the grip force baseline and the modulation, and analyzed each individually, in this experiment we analyzed the overall grip force. The overall grip force is comprised of a predictive and a reactive component.

*Figure 9* shows the ratios for each of the tactor displacement gains in the first and last probing movements, averaged across all the participants in the free exploration group (Group 2). To clarify, this analysis was performed only on Group 2, as Group 1 probed each of the force fields only once, and therefore a comparison between early and late probing movements was not possible. Consistent with previous studies (*Leib et al., 2015*; *Leib et al., 2018*), in trials with no skin-stretch (that is, tactor displacement gain of 0 mm/m), participants decreased their grip force as they formed a representation of the force field that they touched. This analysis revealed that the *peak grip force-peak load force ratio* increased as the tactor displacement gain increased, for both the second (gentle rise) and seventh (steep rise) probes. These effects were statistically significant (rm-General Linear Model, main effect of 'gain': $F_{(1,8)} = 29.07$, $p = 0.0007$; main effect of 'probing movement': $F_{(1,8)} = 17.31$, $p = 0.0032$; interaction between 'gain' and 'probing movement' variables: $F_{(1,8)} = 10.53$, $p = 0.0118$). The *peak grip force-peak load force ratio* for the first probing movement also increased with tactor displacement gain. This increase was much gentler than for the last probing movement, but surprisingly, it was significantly greater than zero (rm-General Linear Model, main effect of 'gain': $F_{(1,8)} = 5.43$, $p = 0.0481$). This was an unexpected result as participants could not predict

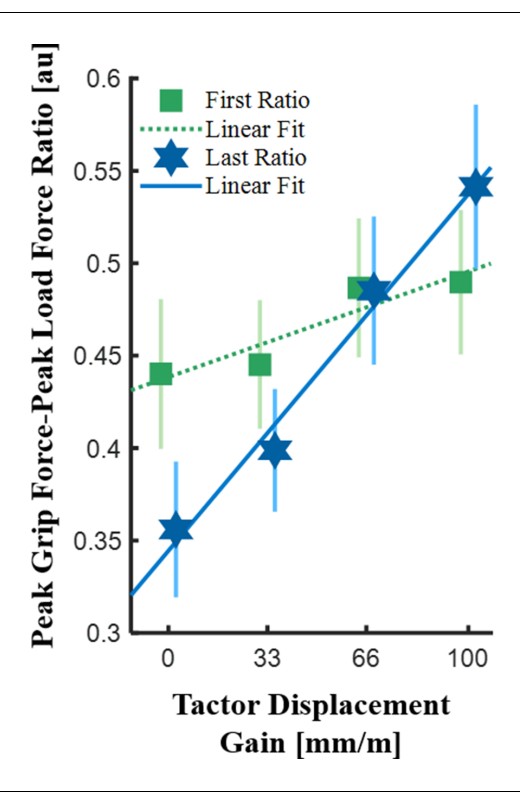

**Figure 9.** Analysis of peak grip force-peak load force ratio from the free exploration group (Group 2). The average ratios in the first (green) and last (blue) probing movements. The markers and vertical lines represent the peak grip force-peak load force ratios averaged across all the participants (N = 9) and their standard errors, respectively. The dotted green and solid blue lines represent the average fitted regression lines.

the tactor displacement gain value in their first interaction with the force field. We therefore posit that this increase in grip force was a result of the reactive grip force response (see Appendix 1).

To conclude, in this experiment, we reproduced the results that adding artificial skin-stretch increases the perceived stiffness (i.e. caused the illusion of touching a harder object) linearly with the tactor displacement gain. We investigated the timescale of the formation of the perceptual effect and found that it was formed immediately upon the exposure to the skin-stretch stimulus. Additionally, we examined the effect of the skin-stretch on the overall grip force when participants were free to probe the force fields as many times as they wished, unlike the structured probing movements they were instructed to make in Experiment 1. This analysis confirmed that, here too, the skin-stretch led to an increase in the overall grip force that developed with subsequent probing interactions with the same elastic force field.

## Discussion

In this study, we examined the effect of artificial tactile feedback on the predictive control of grip force and the perception of stiffness. Previous findings have reported that adding artificial skin-stretch to kinesthetic force feedback causes the perceptual illusion of touching a harder object (*Quek et al., 2013*; *Quek et al., 2014*). Our results suggest that artificial skin-stretch also affected the prediction of the load force applied by the object, and consequentially led to a gradual increase in the predictive modulation of grip force in anticipation of the load force. Additionally, we discovered that the perceptual illusion was formed immediately upon first contact with the object, and did not depend on the slower development of the predictive control of grip force. Previous studies have investigated the contribution of tactile information to the predictive grip force control. Many of these studies focused on patients with impairments (*Nowak and Hermsdörfer, 2003*; *Nowak et al., 2003*; *Witney et al., 2004*) or used digital anesthesia (*Nowak et al., 2001*; *Augurelle et al., 2003*), and concluded that impaired cutaneous feedback increased the overall grip force, but did not affect the modulation of grip force in anticipation of load force. Our study is the first to address this issue in an intact motor system, using artificial cutaneous information. Our results showed that artificial skin-stretch increased the predictive grip force by increasing both the baseline grip force and the modulation in anticipation of the predicted load force. In addition, we determined that the size of this effect was linear with the amount of stretch. Interestingly, the timescales of the evolution of these two components were different – the baseline grip force increased immediately together with the perceptual illusion, and the modulation evolved gradually.

Which aspect of the internal representation caused the increase in the predictive modulation of grip force? This current study cannot provide an answer to this question, but we can raise several possible explanations. Previous studies have determined that artificial skin-stretch causes an increase in the perceived load force (*Gleeson et al., 2009*; *Matsui et al., 2013*; *Quek et al., 2014*). One possibility is therefore that a similar increase in estimated load force can affect the predictive modulation. Alternatively, the estimation of the force field stiffness (*Leib et al., 2016*) could have been updated directly. Another possible explanation is that the participants may have interpreted the additional tactile stimuli as indicative of a more slippery contact surface at the interface with the fingers (*Johansson and Westling, 1984*; *Flanagan and Wing, 1990*), hence leading to an increase in the grip force which is entirely unrelated to the perceptual illusion of a harder object. Moreover, we observed a reactive reduction of the grip force around the peak load force in response to the stretch stimulation (which we interpret with caution - see Appendix 1). This reduction increases the risk of slippage which could be compensated for by increasing both components of the predictive grip force (that is, the baseline and the modulation) that are entirely unrelated to the mechanical properties of the interface with the finger.

Previous studies have shown that to reduce the risk of slippage, participants initially apply excessive grip force by applying a safety margin beyond the sufficient amount of grip force. However, with repeated interactions with the environment, the baseline grip force lessens and adjusts to the expected load force (*Leib et al., 2015*; *Leib et al., 2018*). Surprisingly, we did not observe this type of decrease in the baseline grip force between the second and seventh probes when stretch was applied, suggesting that the participants did not reduce their safety margin, even after the predictive modulation was updated. A possible explanation is that the stretch stimulation may have increased uncertainty about the cutaneous information, and the larger was the stretch the larger was

the uncertainty. This is consistent with the increased JND that we observed for some of the participants with higher levels of tactor displacement gain. Moreover, a recent study showed that grip forces are increased due to variability in the environment (*Hadjiosif and Smith, 2015*). Additionally, uncertainty favors increase in impedance (*Franklin et al., 2003*; *Tee et al., 2010*) which may also be associated with a rapid increase in grip force (*Tsuji et al., 1995*). Therefore, increased uncertainty could have prevented the participants from reducing their baseline grip force.

Consistently with previous studies (*Vega-Bermudez and Johnson, 2004*; *Quek et al., 2013*; *Quek et al., 2014*; *Park et al., 2018*), we found that adding artificial tactile feedback augmented the perceived stiffness, and that the augmentation was a linear function of the tactor displacement gain. Our findings show that this perceptual illusion is immediate, and does not depend on repeated interactions with the force field. This suggests that the perceptual illusion is a result of a high-level cognitive process, independent of the gradual development of the increase in the predictive modulation of grip force. This is consistent with recent findings regarding the fast timescales of explicit cognitive strategies in motor adaptation, compared to the slower implicit adaptation mechanisms (*McDougle et al., 2015*). In fact, the immediate perceptual effect observed in Experiment 2 was evident after the participants were exposed to exactly the same amount of information as the participants in Experiment 1 had when they planned their control of grip force in the second stretch-catch probe. In the second stretch-catch probe, only the baseline grip force component, which reflects uncertainty, increased due to the skin-stretch, whereas the modulation, which reflects the planning of the grip force in anticipation of the predicted load force, changed gradually. Conversely, the perceptual effect after one probe was similar to the perceptual effect after free exploration, and was linearly related to the amount of stretch.

There were two groups in Experiment 2; one which probed each force field a single time, and one that made as many probing movements as they wished (free exploration). Both the effect size (represented by the PSE value) and the sensitivity (represented by the JND value) were similar in both groups of Experiment 2. We expected to observe larger PSE values in the free exploration group, as serial interactions are often combined as a weighted average (*Metzger et al., 2018*). Additionally, we also expected that as multiple interactions with the force field supply more information, this would lower the variability of the estimations of stiffness, and therefore lead to smaller JND values (*Ernst and Banks, 2002*). Our findings, which are in contrast with these expectations, suggest that the skin-stretch stimulus is very strong, and is well above the discrimination thresholds (*Gescheider, 2013*). This offers an explanation for the many compelling effects of tactile stimulation found in a variety of applications.

Our results indicate that the perceptual illusion and the update of the predictive modulation of grip force in anticipation of the load force did not share a similar timescale. However, we cannot determine whether they shared a different stiffness estimation mechanism (*Leib et al., 2016*). Many studies have reported that the motor system is robust to perceptual illusions (*Goodale and Milner, 1992*; *Aglioti et al., 1995*; *Flanagan and Beltzner, 2000*; *Ganel and Goodale, 2003*; *Leib et al., 2015*; *Leib et al., 2018*), but other studies found a lack of dissociation (*Franz et al., 2000*; *Jackson and Shaw, 2000*; *Bruno, 2001*; *Buckingham and Goodale, 2010*). Our findings are similar to those reported by *Flanagan and Beltzner (2000)* in terms of timescales – they found that the perceptual size-weight illusion was immediate and persistent, whereas the grip force control changed over repeated grasping of the objects. However, the nature of the change in the grip force was different - in *Flanagan and Beltzner (2000)* the grip force matched the perception in the first grasping, and was then adjusted to the correct weight. In our case, on the other hand, the predictive modulation of grip force slowly increased over repeated interactions. In general, the debate regarding the existence of dissociation between perception and action remains open.

Understanding the dissociable effects of kinesthetic and tactile cues on grip force control and the integration between these modalities is critical for enhancing feedforward and feedback models, and for multisensory integration. In this study, we conducted two experiments which allowed for the characterization of the effect of artificial skin-stretch on the predictive component of grip force control and the timescale of the formation of perceptual effects throughout subsequent probing movements. Our findings suggest that skin-stretch contributed to an increase in the amplitude of the predictive grip force adjustment in anticipation of the load force. Additionally, the perceptual illusion appears to have been formed immediately upon the first contact with the object, independently of the gradual development of the predictive grip force control. These results are important for

understanding the remarkable human ability to gracefully manipulate a variety of objects manually without breaking or dropping them, and for developing new technologies for simulating the human sense of touch.

## Materials and methods

### Ethics statement

All the participants of the experiments signed a written informed consent form. Prior to signing the form, they read the information in a printed form, and heard an explanation by the experimenter. The procedures and the consent form were approved by the Human Subjects Research Committee of Ben Gurion University of the Negev, Be'er-Sheva, Israel, approval number 1283–1, dated from July 6th, 2015.

### Experimental setup

#### Skin-stretch device

The goal of this study was to understand the contribution of adding artificial skin-stretch to kinesthetic force feedback for modulating grip force and forming stiffness perception. To achieve this goal, we disengaged the natural relationship between kinesthetic and tactile information sources using a custom-built 1 DoF skin-stretch device (*Figure 1*) which allowed us to apply different levels of skin-stretch to the user's fingers. Participants placed their thumb and index fingers on the designated locations, and the device stretched the skin of the finger pads by tactor displacement. Our device was based on the design in *Quek et al. (2014)*, with several modifications. The device was equipped with a DC micro motor (Faulhaber, series 1516-SR), a spur gearhead (Faulhaber, series 15/8 with gear ratio of 76:1), an encoder (Faulhaber, series IE2-1024), and an analog motion controller (Faulhaber, series MCDC 3002). We integrated a force sensor (ATI, Nano17) to measure the applied grip force, which was mounted on the lower end of the device so that participants did not place their fingers directly above the force sensor. The left side of the outer shell consisted of a 'door' with an axis on its upper end, and a cylindrical protrusion facing the force sensor. When the device was held with the index finger and thumb on the aperture, the protrusion pressed the force sensor and the relative grip force was measured. Because of the colocation of the tactor movement mechanism and the ideal placement of the force sensor, we could not measure the grip force directly. However, the division of the grip force between the tactor and the aperture, and the placement of the force sensor at a distant location allowed, through the law of conservation of angular momentum, to measure a downscaled version of the grip force that the participants applied on the aperture of the skin-stretch device. This means that we measured trends in the grip force that were linearly related to the actual grip force that the participants applied (see Appendix 2). The weight of the skin-stretch device was approximately 200g and was compensated by a weight that was mounted on the haptic device (*Taati et al., 2008*). The purpose of the weight compensation was to reduce the effect of gravity on the load force. We also examined the magnitude of the inertia force on the load force. Meaning, the participants moved the tactile device at a certain acceleration rate, and we were interested in ensuring that this did not cause them to feel additional forces. We calculated the inertia force and found it to be negligible in comparison to the load force. The average maximum acceleration value was $0.2392\frac{m}{s^2}$, with a standard deviation of $0.0924\frac{m}{s^2}$. The weight of the tactile device was approximately $200g$ and therefore the inertia force is $\approx 0.05\,N$. The range of the load force during the experiment was $\approx 2-4N$, which is two orders of magnitude larger than the inertia force. Hence, both the gravity force and the inertia force are negligible in comparison to the load force.

### System setup

Participants sat in front of the virtual reality system and held the skin-stretch device that was mounted on a PHANTOM Premium 1.5 haptic device (Geomagic) with the index finger and thumb of their dominant right hand (*Figure 1*) and looked at a semi-silvered mirror showing the projection of an LCD screen placed horizontally above it. An opaque screen was fixed under the mirror to block the view of the hand. During the experiment, participants wore noise-cancelling headphones (Bose QuietComfort 35 QC35) to eliminate auditory cues from the motor. We used the haptic device to

apply a kinesthetic virtual elastic force field in the vertical upward direction (*y* in **Figure 1**). The kinesthetic force and the skin-stretch were applied along the same direction, and only after participants were in contact with the force field. Participants could make and break contact with the force field by moving their hand along the vertical axis, and did not experience any force or stretch while moving along the positive half of the y-axis. While moving along the negative half of the axis, participants experienced force that was proportional to the amount of penetration distance (**Video 1**).

$$f_{kin} = \begin{cases} -k \cdot y, & y \leq 0 \\ 0, & y > 0 \end{cases} \qquad (1)$$

where k [N/m] is the stiffness, and y [m] is the penetration distance into the virtual force field. We used the skin-stretch device to apply tactile stimuli by means of tactor displacement:

$$y_{tactor} = \begin{cases} -g \cdot y, & y \leq 0 \\ 0, & y > 0 \end{cases} \qquad (2)$$

where g [mm/m] is the tactor displacement gain, and y [m] is the same penetration distance into the force field.

## Protocol

In all experiments, we asked participants to probe pairs of virtual elastic *standard* and *comparison* force fields and indicate which force field had a higher level of stiffness. Each force field was indicated to the participants as a screen color that was either red or blue, which we defined randomly prior to the experiment. After interaction with the force fields and after choosing which one had a higher level of stiffness, participants pressed a keyboard key that corresponded to the screen color of the stiffer force field. For each trial, the stiffness of the *comparison* force field was also pseudo-randomly chosen prior to the experiment. During interaction with the *comparison* force field, only kinesthetic force was applied. Interaction with the *standard* force field was augmented with one of four different levels of tactor displacement gain (0, 33, 66, and 100 mm/m), in addition to the applied load force. After choosing the force field that was perceived as stiffer, the screen became grey, and participants initiated the next trial by raising the end of the robotic device. There was no visual feedback along the vertical axis during the experiment. To avoid force saturation of the robotic device, we used an auditory alert if the penetration into the force field exceeded 40 mm. To avoid saturation of the skin-stretch device, we limited the range of the tactor-displacement gains that we investigated to 100 mm/m. This ensured that during typical interactions with the force fields, the tactor did not reach the aperture of the device body, and caused as few saturation cases as possible. To become familiarized with the experimental setup and to ensure that participants understood how to grasp the device, a training session was carried out at the beginning of each experiment. During training, we provided feedback to participants about their responses.

## Experiment 1

Ten participants (seven females, 22–26 years old) participated in the experiment after signing an informed consent form as approved by the Human Subject Research Committee of Ben Gurion University of the Negev, Be'er-Sheva, Israel.

The experiment focused on the predictive components of grip force control due to the skin-stretch stimulus. The *standard* force field had a constant stiffness value of 85 N/m, and during interaction with this force field, the tactor displacement gain was either 0, 33, 66, or 100 mm/m. We asked participants to distinguish between pairs of force fields, but because in this

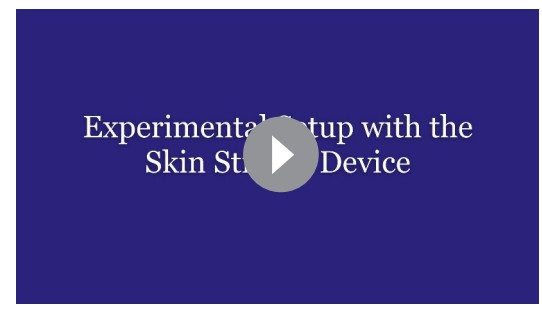

**Video 1.** Experimental system with the skin-stretch device. This video clip demonstrates the way in which participants performed the experiments; what they did with their right hand and what they saw on the screen. In addition, the video clip includes a zoom in on the skin-stretch device without the participant's hand. The zoom in allowed us to show the tactor displacement during interaction with the elastic force field.
https://elifesciences.org/articles/52653#video1

experiment we were not analyzing stiffness perception, we only used three stiffness values for the *comparison* force field (40, 85, and 130 N/m). We asked participants to perform eight discrete movements into each of the two force fields. We only counted successful movements, namely those that started and ended outside the elastic field and extended at least 20 mm into the force field, and that were completed within 300 msec. We presented a counter of successful movements to the user. After eight probing movements with the first force field, the field automatically switched to the second force field. When participants had performed eight probing movements into the second force field, the screen automatically became white and participants had to choose the force field that they perceived as stiffer.

The experiment consisted of 132 trials, 24 training trials and 108 test trials, and was divided into two sessions of 66 trials that were completed in two days. Training consisted of four repetitions of the three levels of stiffness for the *comparison* force field, and a 0 mm/m tactor displacement gain for the *standard* force field (participants performed a training set at the beginning of each day). The protocol for the test study consisted of nine repetitions of trials with 12 pairs of force fields (three levels of stiffness for the *comparison* force field and four levels of tactor displacement gain for the standard force field). To dissociate the contribution of the feedforward grip force adjustments, we incorporated stretch-catch probes where we maintained the load force but unexpectedly omitted the skin-stretch (*Hermsdörfer and Blankenfeld, 2008*). In these probing movements, participants expected to feel the skin-stretch, as they did in the previous probing movement into the same force field. They therefore predictively adjusted their grip force in anticipation of this stimulus. As it was a stretch-catch probe, there was no skin-stretch to apply a reactive grip force to. In six of the nine trials, we introduced stretch-catch probes in either the second or in the seventh movement, in which we maintained the load force but unexpectedly omitted the skin-stretch. In three of the six trials, the stretch-catch probe was the second probing movement of the total of eight probes, and in the other three, it was the seventh probing movement. Overall, 3.125% of the probes were stretch-catch probes. The idea was to use the predictive grip force adjustment to assess the effect of the added tactile stimulation on the anticipated elastic force field during early and late interaction with each force field. For early interaction, we chose to introduce stretch-catch probes in the second and not the first probing movement since it would have been meaningless to omit the tactile stimulus before participants felt it at least once. All the different conditions were presented in a pseudorandom predetermined order.

## Experiment 2

We designed the second experiment to understand whether the previously reported bias in stiffness perception is due to high-level cognitive processing of the artificial stretch stimulations and occurs immediately, or if it develops with the accumulation of a change in the predictive modulation of the grip force, and potentially as a result of this updated representation. To address this issue, we used a stiffness discrimination task, and examined the dependence of the effect of adding artificial skin-stretch to a kinesthetic load force on the number of serial interactions with the elastic force field. A total of N = 20 participants (12 females, 22–26 years old) participated in the experiment after signing an informed consent form as approved by the Human Subject Research Committee of Ben Gurion University of the Negev, Be'er-Sheva, Israel. We divided the participants into two groups: Group 1 ($N_1$ = 10, six females) who were only allowed to probe each force field once, and Group 2 ($N_2$ = 10, six females) who were free to probe the elastic force fields as they wished.

In both groups, the *standard* force field had a constant stiffness value of 85 N/m and the tactor displacement gain was either 0, 33, 66, or 100 mm/m. We chose the *comparison* force field stiffness from a range of values between 40–130 N/m spaced at intervals of 10 N/m. To switch between the two force fields, participants had to exceed at least 30 mm beyond the boundary of the force field by raising the robotic device vertically. Training consisted of two repetitions of the 10 levels of stiffness for the *comparison* force field, and a 0 mm/m tactor displacement gain for the *standard* force field. During training, skin-stretch was never applied, that is, all the *standard* and *comparison* fields of the training trials had a gain of 0 mm/m. The test trials were comprised of 40 different force fields pairs (10 *comparison* stiffness levels and four *standard* tactor displacement gains) which were repeated eight times throughout the test session. All the different conditions were presented in a pseudorandom and predetermined order.

### Group 1

Participants could probe the force fields and switch between them just once. Hence, participants had touched each force field only once before they made a decision about the stiffer force field, similarly to *Di Luca et al. (2011)*. The experiment was completed in one session and comprised of 20 training trials (not analyzed) and 320 test trials.

### Group 2

Participants used free exploration. They could probe the force fields and switch between them as many times as they desired to make their decision. Individual probing trajectories were also unrestricted and they could choose the way they moved; for example, in a rhythmic or discrete manner. The experiment was completed in two sessions, each of which began with 20 training trials (which were excluded from the analyses), followed by 180 test trials.

## Data analysis

Sample size: we expected the effects in our study to be similar to the previous studies of the effect of skin-stretch on perception of stiffness (*Quek et al., 2014*), and grip force modulation during interaction with elastic force fields (*Leib et al., 2015*). Therefore, we chose N = 10 for each group in our study.

Statistical analysis: we used the Lilliefors test (*Lilliefors, 1967*) to assure that in most of the cases, the dependent variables came from normal distributions within each group. We used a 0.05 threshold for statistical significance. Other statistical analyses are presented in individual experiments.

In the following analyses of grip force control, for each of the dependent variables separately, we used a General Linear Model with a continuous factor (tactor displacement gain, df = 1), a categorical factor (movement number, df = $N_m$-1, where $N_m$ is the number of the tested probing movements in each analysis), a random factor (participants, df = $N_p$-1, where $N_p$ is the number of the participants in each experimental group), and their respective interactions. In the analysis of the stretch-catch probes, the number of movements in each model was $N_m$ = 3 for the zero gain case (probing movements 1,2, and 7) and $N_m$ = 2 for the other gains (probing movements 2 and 7). In the analysis of the evolution of grip force during all the probing movements within a trial the number of movements was $N_m$ = 7 (all the probing movements were included in the model except from the last one). In the analysis of the perception (separately for PSE and JND), we used repeated measures regression with the tactor displacement gain as the independent variable (df = 1), participants as a random factor (df = $N_p$-1), and their interaction.

## Experiment 1

We recorded the grip force data and filtered it using the MATLAB function *filtfilt* with a second-order Butterworth low-pass filter, with a cutoff frequency of 12 Hz, resulting in a fourth-order filter, with a cutoff frequency of 9.62 Hz. For each trial, we examined the trajectories of the load force that were generated by the robotic device, the grip force that participants applied, and the tactor displacement. We separated the probing movements and analyzed the grip force trajectories associated with the applied load force and tactor displacement. We identified the start and the end of each probing movement using the load force signal. The load force was equal to zero when the vertical position of the end of the haptic device was greater than zero, and increased as a function of the penetration once the participants crossed the boundary of the elastic force field. Analyses were only run on data collected during interactions with the *standard* force field because it had a constant stiffness level. Rare cases in which participants did not hold the device adequately and released their grip were excluded from the analysis since we could not measure the applied grip force in these trials.

To isolate the predictive component of grip force control due to the skin-stretch stimulus, we performed three analyses. In the first two we only examined the applied grip force from probing movements with stretch-catch probes (second and seventh probes). These two enabled the analysis of only the second and seventh probes. Therefore, to examine the evolution of the predictive grip force over repeated interaction with the elastic force field, we used the approach described in *White et al. (2011)* and *Kuling et al. (2019)* to analyze all the probing movements. This approach

makes use of the grip force that participants applied before the movement initiation to calculate the grip force baseline and to estimate the maximum grip force.

In the first two analyses, for a given trial, we examined the load force applied during interactions with the force field, and the applied grip force. We were interested in evaluating the change in grip force control between the initial and final probing movements of each interaction. To do so, in trials without skin-stretch we compared the first, second, and seventh probing movements. In trials with skin-stretch, we analyzed the second and seventh stretch-catch probes separately, and compared between them. To quantify the control of grip force during repeated exposure to the elastic force field with and without artificial skin-stretch, we performed two analyses: (1) *peak grip force-peak load force ratio* and (2) *grip force-load force regression*.

## Peak grip force-peak load force ratio analysis

We calculated the peak grip force-peak load force ratio of the second and seventh probing movements for each level of tactor displacement gain. We hypothesized that during the second probing movement, participants would have little information about the tactor displacement gain that was used for the *standard* force field, whereas in the seventh probing movement, participants would already have information about the stimulus and likely have formed an estimation of it. To test the significance of the change in the applied grip force due to tactor displacement gain and between probing movements, we used a repeated-measures General Linear Model using the *anovan* MAT-LAB function. The independent variables were the tactor displacement gain (continuous, df = 1), the probing movement (categorical, two levels - second or seventh, df = 1), and participants (random, df = 9). The model also included interactions between the independent variables.

## Grip force-load force regression analysis

To separate the modulation of grip force in anticipation of the load force from the baseline grip force control, we analyzed the grip force-load force regression, similar to the analysis in *Flanagan and Wing (1990)*; *Nowak et al. (2002)*; *Leib et al. (2015)*. We fit a two degrees-of-freedom regression line (slope and intercept) to the trajectory in the grip force-load force plane for each of the stretch-catch probes (second and seventh probing movements), as well as for trials with a tactor displacement gain of 0 mm/m. The intercept represents the grip force baseline; that is, the amount of grip force that participants applied when no external load force was applied by the haptic device. The slope of the regression represents the modulation of the grip force in anticipation of the load force. Rare cases in which the slope was negative (indicating a large phase shift between the load force and the grip force) were excluded from the analysis. We fit a repeated-measures General Linear Model to each the slope and the intercept as the dependent variables in two separate statistical analyses, using the anovan MATLAB function. The independent variables were the tactor displacement gain (continuous, df = 1), the probing movement (categorical, two levels - second or seventh, df = 1), and participants (random, df = 9). The model also included interactions between the independent variables.

## Grip force baseline and prediction of the intended peak grip force in all the probing movements in each trial

We were interested in quantifying the evolution of the different components of the predictive grip force control in all the successful probing movements within a trial, rather than in just the stretch-catch probes. To accomplish this, we followed the approach proposed in *White et al. (2011)*; *Kuling et al. (2019)*.

The *grip force baseline* was calculated as the grip force between two consecutive probing movements. We found the time between probing movements (not in contact with the elastic force field) at which the change in the grip force was minimal: we calculated the grip force derivative and detected when it was close to zero (less than 0.1 [N/s]). We then calculated the average grip force value along 200 ms before this time as the *grip force baseline*. We calculated the *grip force baseline* for each participant, for all four tactor displacement gains (0, 33, 66, and 100 mm/m), for each of the first seven probing movements. We omitted the last probing movement from the analysis because in this movement the participants were getting ready to finish the trial and performed irregular probing movements.

## A multivariate regression model for the intended peak grip force

We adopted the analysis in *White et al. (2011)*; *Kuling et al. (2019)*, where grip force signals were mechanically disturbed by the impact loads following collisions. Similarly, in our study, the grip force measurement is distorted because of the movement of the tactors during the interaction with the force field (see Appendix 1). It is well-documented that an increase in the grip force and its rate of change precede an increase in the load force during contact with dynamic objects, and therefore, a multivariate linear regression model can be used to predict the intended peak grip force from the grip force and the grip force rate at the initial contact with the object (*White et al., 2011*; *Kuling et al., 2019*). Such analysis allows for extracting the predictive component of grip force control from probing movements in which skin-stretch was applied by only using information before the stretch stimulation.

We used data from all the stretch-catch probes to train a multiple regression model that predicts the peak grip force based on the grip force and the grip force rate at contact. The *grip force at contact* was calculated as the grip force at the first sample in contact with the elastic force field. The grip force rate at contact was calculated with a backward difference approximation of the derivative of the grip force signal with respect to time. The resulting model was:

$$GF_{peak} = 1.14 GF_{contact} + 0.06 \frac{dGF_{contact}}{dt} + 0.1 \qquad (3)$$

with $R^2=0.732$, and both independent variables contributing significantly to the prediction of the *intended peak grip force* (p<0.001). The coefficients we received are similar to those described in *Kuling et al. (2019)*; *White et al. (2011)*, and varied only minimally when we fitted the model separately for the second and seventh probing movements or across different tactor displacement gains. Then we used this model to estimate the *intended peak grip force* for all the probing movements.

The *intended peak grip force* is determined by the *grip force baseline* as well as the modulation of grip force in anticipation of load force. To get a better assessment of the evolution of these two very different components of predictive grip force control, and to compare the results of this analysis to the *grip force-load force regression* analysis that we performed in the stretch-catch probes, we further separated this analysis to *grip force at contact* and *grip force modulation*. We calculated the *grip force modulation* by subtracting the *grip force at contact* from the *intended peak grip force*. For the ideal case of a grip force that linearly depends on load force, the *grip force at contact* is identical to the *intercept* of the *grip force-load force regression*. The *grip force modulation* is related to the *slope* of the regression, but would have to be divided by the *peak load force* to have identical units and meaning. However, this would require the use of a value that occurred during the movement, once the skin-stretch was applied, which we were interested in avoiding. Therefore, we analyzed the *grip force modulation* without information about the amount of load force in each interaction, with a metric that has units of grip force instead of arbitrary units. While it would generally be difficult to interpret this metric based on interactions with elastic objects, in our case the extent of all the probing movements was similar. Nevertheless, some qualitative differences between the slope and the *grip force modulation* measurements are likely.

We fitted a repeated-measures General Linear Model to each the *grip force baseline, intended peak grip force, grip force at contact,* and *grip force modulation* as the dependent variables in four separate statistical analyses, using the anovan MATLAB function. The independent variables were the tactor displacement gain (continuous, df = 1), the probing movement (categorical, seven levels, df = 6), and participants (random, df = 9). The model also included interactions between the independent variables.

## Experiment 2

### Perception

For each participant, we fitted psychometric curves for the probability of responding that the comparison force field was stiffer as a function of the difference between the stiffness of the comparison and the standard force field using the Psignifit toolbox 2.5.6 (*Wichmann and Hill, 2001*). We repeated this procedure for every level of tactor displacement gain. To assess the effect of artificial skin-stretch on perception of stiffness, we computed the point of subjective equality (PSE) and the just noticeable difference (JND) of each psychometric curve. The PSE indicates the difference in

stiffness levels at which the probability of responding that the comparison force field had a higher level of stiffness was 0.5. A positive PSE value; that is a rightward shift of the psychometric curve, represents overestimation of the standard force field, and a negative PSE value; that is a leftward shift indicates an underestimation. The JND was defined to be half the difference between the comparison stiffness levels corresponding to the 0.75 and the 0.25 probabilities of responding that the comparison force field had a higher level of stiffness. The JND indicates the sensitivity of the participant to small differences between the stiffness levels of the two force fields.

For each of the groups separately, after extracting four PSE and four JND values for each participant (for each level of tactor displacement gain), we fitted regression lines to these values. To test the significance of the change in PSE and JND values due to tactor displacement gain, we used a repeated-measures regression with the *anovan* MATLAB function. The independent variables were the tactor displacement gain (continuous, df = 1), and participants (random, group 1: df = 9, group 2: df = 8). The model also included interactions between the independent variables. In addition, to test if there was a difference between the effect of the tactor displacement gain on the two groups, we used a nested repeated-measures General Linear Model. The independent variables were the tactor displacement gain (continuous, df = 1), the group number (categorical, two groups – first or second, df = 1), and participants (random, df = 18). The model also included interactions between independent variables.

### Action - Peak grip force–peak load force ratio

To reproduce the results of the change in the predictive grip force between early and late interactions with the elastic force field (observed in Experiment 1) we computed the *peak grip force-peak load force ratio* in Group 2 of Experiment 2. Unlike in Experiment 1, in which we only analyzed the grip force-load force trajectories of the stretch-catch probes, in Experiment 2 we examined the effect of the additional skin-stretch on the overall grip force trajectories. The grip force trajectories had a non-uniform peaked pattern that appeared predominantly in trials in which skin-stretch was applied. Therefore, we used the MATLAB function *findpeaks* to identify the grip force and load force peaks, and then manually corrected them by visual examination. Trials in which participants probed the force field only once were excluded from the analysis since we could not compare between the first and the last probes. To test the significance of the change in the applied grip force due to tactor displacement gain and between the first and the last probing movements, we used a repeated-measures General Linear Model using the *anovan* MATLAB function. The independent variables were the tactor displacement gain (continuous, df = 1), the probing movement (categorical, two levels – first or last, df = 1), and participants (random, df = 8). The model also included interactions between the independent variables.

## Acknowledgements

The authors thank Amit Milstein, Zhan Fan Quek, and Eli Peretz for their help and advice in designing and building the skin-stretch device, and Guy Avraham for valuable comments on the manuscript. This study is supported by the Binational United-States Israel Science Foundation (grant no. 2016850), by the National Science Foundation (grant no. 1632259), by the Israeli Science Foundation (grant 823/15), by the Ministry of Science and Technology (Israel-Italy virtual lab on 'Artificial Somatosensation for Humans and Humanoids'), and by the Helmsley Charitable Trust through the Agricultural, Biological and Cognitive Robotics Initiative of Ben-Gurion University of Negev, Israel. Mor Farajian was supported by the Tzin Fellowship. Hanna Kossowsky was supported by the Lachish Fellowship.

## Additional information

### Funding

| Funder | Grant reference number | Author |
| --- | --- | --- |
| Israel Science Foundation | 823/15 | Ilana Nisky |
| National Science Foundation | 1632259 | Ferdinando A Mussa-Ivaldi |

| United States-Israel Binational Science Foundation | 2016850 | Ilana Nisky |
| Ben-Gurion University of the Negev | ABC Robotics Initiative | Ilana Nisky |
| Ministry of Science and Technology, Israel | Israel-Italy virtual lab on 'Artificial Somatosensation for Humans and Humanoids' | Ilana Nisky |
| Ben-Gurion University of the Negev | Tzin Fellowship | Mor Farajian |
| Ben-Gurion University of the Negev | Lachish Fellowship | Hanna Kossowsky |

The funders had no role in study design, data collection and interpretation, or the decision to submit the work for publication.

## Author contributions

Mor Farajian, Conceptualization, Data curation, Software, Formal analysis, Validation, Investigation, Visualization, Methodology, Writing - original draft, Writing - review and editing; Raz Leib, Software, Formal analysis, Investigation, Visualization, Methodology, Writing - original draft, Writing - review and editing; Hanna Kossowsky, Investigation, Visualization, Methodology, Writing - original draft, Writing - review and editing; Tomer Zaidenberg, Software, Methodology, Building the first prototype of the skin-stretch device and its control software; Ferdinando A Mussa-Ivaldi, Conceptualization, Funding acquisition, Investigation, Methodology, Writing - review and editing; Ilana Nisky, Conceptualization, Resources, Formal analysis, Supervision, Funding acquisition, Validation, Investigation, Visualization, Methodology, Writing - original draft, Project administration, Writing - review and editing

## Author ORCIDs

Mor Farajian https://orcid.org/0000-0003-0545-563X
Raz Leib https://orcid.org/0000-0002-3940-2651
Hanna Kossowsky https://orcid.org/0000-0002-5058-2382
Ferdinando A Mussa-Ivaldi http://orcid.org/0000-0001-5343-7052
Ilana Nisky https://orcid.org/0000-0003-4128-9771

## Ethics

Human subjects: All the participants of the experiments signed a written informed consent form. Prior to signing the form, they read the information in a printed form, and heard an explanation by the experimenter. The procedures and the consent form were approved by the Human Subjects Research Committee of Ben Gurion University of the Negev, Be'er-Sheva, Israel, approval number 1283-1, dated from July 6th, 2015.

## Decision letter and Author response

Decision letter https://doi.org/10.7554/eLife.52653.sa1
Author response https://doi.org/10.7554/eLife.52653.sa2

# Additional files

## Supplementary files

• Transparent reporting form

## Data availability

Our analysis code and data is available via GitHub, at https://github.com/bgu-SkinStretch/Farajian_et_al2020 (copy archived at https://github.com/elifesciences-publications/Farajian_et_al2020).

TheSolidWorks parts of the skin-stretch device will be provided at request from the corresponding author.

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

## Appendix 1

### Introduction

In Experiment 1, in addition to analyzing the predictive control of the grip force that is presented in the main paper, we wanted to isolate the effects of artificial skin-stretch on the reactive (feedback) control of grip force during *active* interaction with an elastic force field. We also designed and performed an additional experiment, in which we identified the reactive response to artificial skin-stretch during maintenance of constant grip force, that is, a *passive* stretch that did not correlate with a movement of the user. However, because of the close proximity between the moving tactor and the measurement of grip force, we suspected that there may be an artifact of the tactor movement in our grip force measurements that could affect calculation of the reactive response in both experiments. Therefore, we conducted two control experiments to quantify the tactor displacement artifact on grip force measurement without human contact. Appendix 1 presents the results of the analyses of the **reactive control of grip force** during both active and passive experiments (Reactive Experiment 1 and Reactive Experiment 2, respectively). Additionally, Appendix 1 displays the results of two control experiments that were conducted with the purpose of **identifying possible experimental device artifacts** in conditions resembling the active and passive experiments (Control Experiment 1 and Control Experiment 2, respectively). We conclude with a brief **discussion** of the results of all the four experiments taken together.

### Reactive experiment 1

#### The effect of skin-stretch on the reactive component of grip force control during active probing of an elastic force field

We quantified the reactive component of grip force control in response to the added skin-stretch of the fingertips, in both early and late probing movements from Experiment 1 of the main paper. First, we identified the onset and end of the second and seventh probes by finding the first and the last sample during the interactions with the force field. Next, we isolated each trajectory from 50 samples before the onset and 50 samples after the end of the interaction. This measure was taken to ensure that we capture all the grip force data. We then divided each grip force trajectory by the peak load force in the same probe, and time-normalized and aligned each trajectory such that 0 and 1 were the onset and end of the contact with the load force, respectively. The next step was to separate the trials according to the tactor displacement gain and average the grip force trajectories from the second and seventh probes (from both regular and stretch-catch probes). To eliminate variability due to baseline grip force level, we subtracted the onset grip force value from each averaged trajectory. Finally, we subtracted the average grip force trajectoryies applied during stretch-catch probes from the average grip force trajectories applied during regular probes. The resulting trajectories representthe reactive grip force responses to the stretch stimulus from early and late exposure. We calculated the early and late reactive responses of each of the participants to each level of tactor displacement gain and averaged them to obtain the average early and late responses across all the participants.

*Appendix 1—figure 1* presents these reactive grip force responses from the second (a) and seventh (b) probes for different tactor displacement gains. Surprisingly, the stretching of the skin caused a reactive decrease in grip force followed by a reactive increase at the end of the probing interaction. Both the decrease and subsequent increase were stronger for larger tactor displacement gains. This pattern of reactive response remained the same between the second and seventh probes, but the differences between the peak decrease and peak increase became larger with repeated interaction with the stretch stimulation (rm-General Linear Model, main effect of 'gain': $F_{(1,9)} = 101.87$, $p<0.0001$; main effect of 'probing movement': $F_{(1,9)} = 7.40$, $p = 0.0236$; interaction between 'gain' and 'probing movement' variables: $F_{(1,9)} = 2.60$, $p = 0.1416$). This pattern of the reactive grip force response is likely

responsible for the irregular peak pattern that we observed in *Figure 2* (Experiment 1) and in *Figure 8* (Experiment 2).

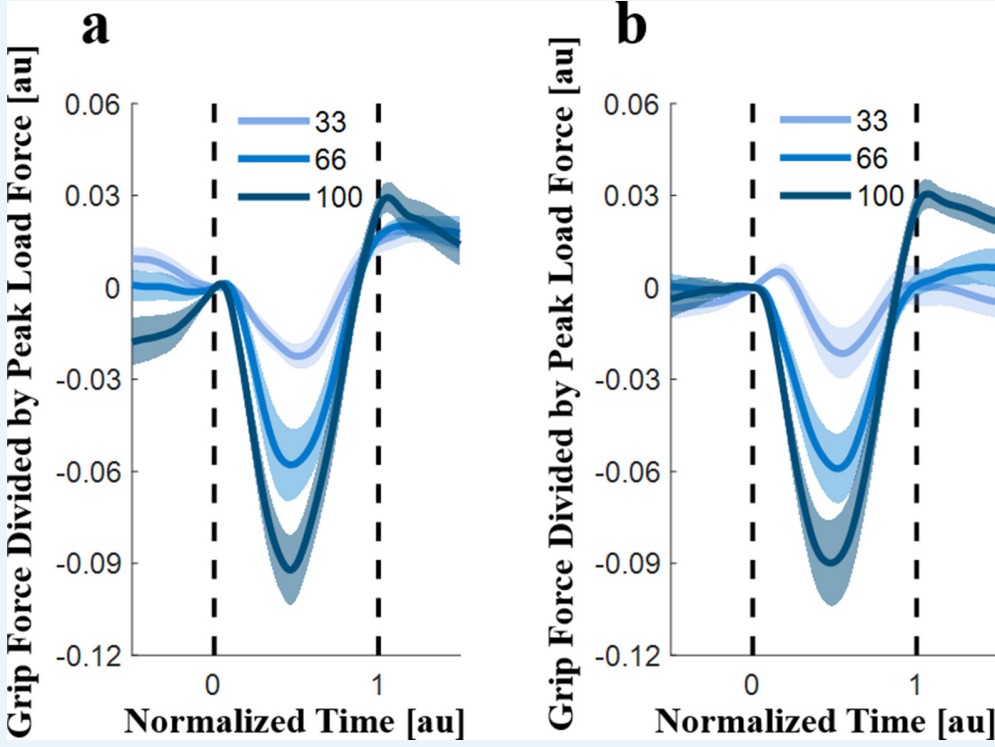

**Appendix 1—figure 1.** Reactive Experiment 1 (active experiment). The reactive grip force response for different levels of tactor displacement gain from the second (**a**) and seventh (**b**) probing movements. The lines represent the mean trajectories across all the participants (N = 10), the shaded regions represent the standard errors, and the black vertical dashed lines represent the time of the interaction with the elastic force field.

## Reactive experiment 2

### The effect of skin-stretch on the reactive component of grip force control during a task of constant grip force maintenance

We tested participants' grip force reactions to different amounts of skin-stretch during a task of maintaining a constant grip force while the position of the skin-stretch device was kept stationary by the Phantom haptic device. Six participants took part in the study after signing an informed consent form, and they were instructed to hold the skin-stretch device with a stable grip force based on visual feedback.

At the beginning of each trial, participants were presented with two bars on the screen. One of them was a green 'target force bar' that indicated a constant target range of grip force that participants were requested to apply. The second bar was red, and its height corresponded to the actual level of grip force applied by the participants. When participants applied grip force on the device, the height of the red bar increased so that its upper border approached the 'target force bar'. When the upper border touched the 'target force bar', the red bar turned blue, and participants were requested to maintain their grip force. Once participants maintained that force for a randomly selected time interval (uniformly distributed between 1 and 4 sec), the tactor started stretching the skin. The trajectory of the displacement of the tactor was $d = g_s A \sin(bt)$, where $A = 3$ and $b = 2\pi$ were chosen to be similar to the average trajectories of the probing movements that the participants of Experiment 1 performed, and gs was chosen to be one of the three tactor displacement gains (33, 66, and 100 mm/m) that we used in this paper. After the end of the tactor movement, the bar

changed the color back to red, the participants released the grip, and this concluded the trial. Each tactor displacement gain was presented in 15 trials throughout the experiment, and the order of presentation of the different tactor displacement gains was randomized, predetermined, and identical for all participants. Each participant repeated this experiment twice with two different levels of target grip force: 1.2 N and 1.6 N. These target grip force levels were both within the grip force range that participants applied in Experiment 1, which varied between 0.5-2 N.

For each participant, we averaged the trajectories of each tactor displacement gain and each target grip force level. *Appendix 1—figure 2* depicts the averaged grip force trajectories of participants separately for each level of target grip force, and for different levels of tactor displacement gain. As in the active experiment, we observed a pattern of a decrease in the grip force followed by an increase toward the end of the stretch. Additionally, we found a similar dependence on the tactor displacement gain – both the decrease and the subsequent increase were greater for larger tactor displacement gains. In addition, a larger target grip force led to a deeper decrease (rm-General Linear Model, main effect of 'gain': $F_{(1,28)} = 26.43$, $p<0.0001$, main effect of 'target grip force': $F_{(1,28)} = 13.44$, $p = 0.0010$). Both reactive experiments taken together suggest that the pattern of the reactive response was not specific to the control of grip force during probing movements, but rather a more general response to the tactile stimulation that we applied in this study.

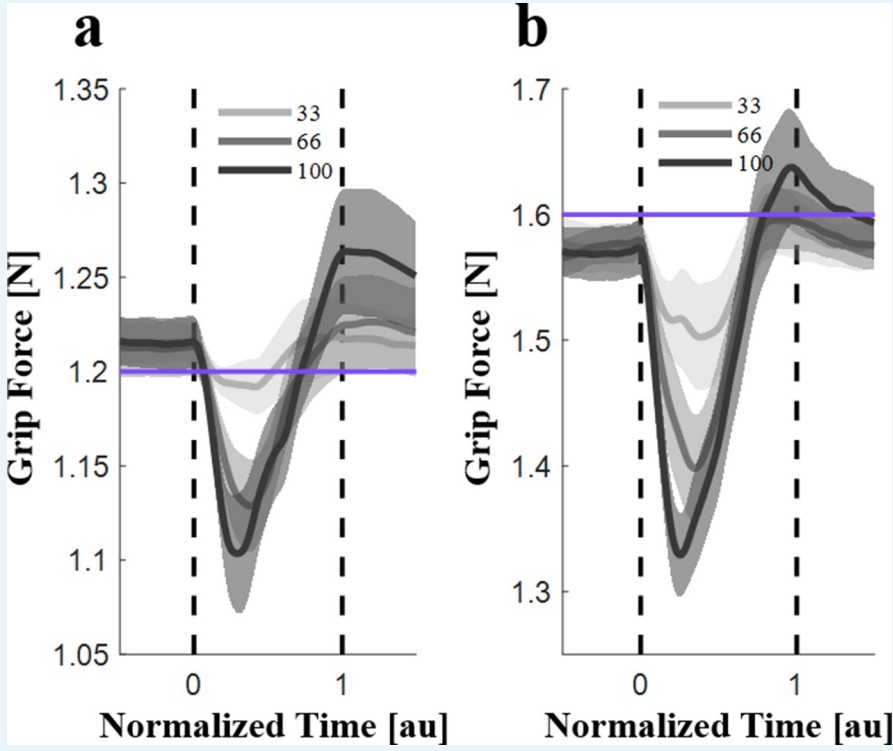

**Appendix 1—figure 2.** Reactive Experiment 2 (passive experiment). Averaged grip force trajectories across all the participants (N = 6) for different levels of tactor displacement gain. Shading represents the standard errors, and the black vertical dashed lines represent the time of the interaction with the stretch. The purple horizontal lines represent the target grip force. (**a**) Target grip force of 1.2N. (**b**) Target grip force of 1.6N.

## Control experiments - artifact characterization experiment

To quantify the effect of tactor movement on the measured grip force without any human input, we conducted two control experiments. In both experiments, we used a padded clamp to apply a constant grip force, and measured the grip force during movement of the tactor.

## Control experiment 1

In this experiment, we reproduced the conditions of Reactive Experiment 1, but without human input for controlling grip force. Therefore, we held the top of the skin-stretch device (in a manner that did not affect the recorded grip force) which was connected to the haptic device, and performed a number of interactions with a force field which had a constant stiffness level of 85 N/m, and different levels of tactor displacement gain. The purpose of this manual interaction was to ensure that we used a human-like probing trajectory similar to the ones that the participants employed during Experiment 1, and that the trajectories of the movement of the tactor would be similar to those that were applied during Experiment 1. The analysis of grip force trajectories was identical to that in Reactive Experiment 1, including time-normalization and alignment of the signals, division by the peak load force, and averaging across repetitions. The grip force was expected to remain constant under the effect of the clamp, and hence, we treated any change in the grip force trajectory in this control experiment as an artifact of tactor movement.

*Appendix 1—figure 3* presents the resulting average grip force artifact trajectories normalized by the peak load force. The average grip force trajectories had a constant pattern: when the tactors started moving, there was a steep decrease in the measured grip force followed by its recovery and then a moderate increase. The size of the artifact in the measured grip force depended on the tactor displacement gain. Comparing the results of this control experiment with the results of the reactive Experiment 1 reveals that the pattern of the artifact trajectory resembles the measured human reactive response, but the magnitude of the artifact is smaller by a factor of 4.

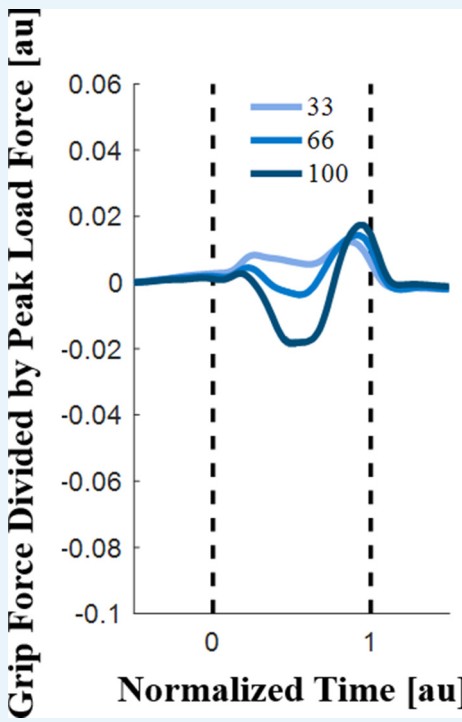

**Appendix 1—figure 3.** Control Experiment 1 (active experiment). Average grip force

artifact trajectories divided by the peak load force. Shading represents the standard errors, and the black vertical dashed lines represent the time of the interaction with the elastic force field.

## Control experiment 2

In this experiment, we applied different amounts of tactor movement without moving the skin-stretch device, and recorded the grip force trajectories. The skin-stretch device was mounted on a haptic device that was maintained stationary, and we used a padded clamp to maintain a stable grip force. After 1-4 s during which the padded clamp maintained a grip force of 1.8N, the tactor was displaced by different amounts. Similarly to Reactive Experiment 2, the trajectory of the displacement of the tactor was $d = g_sAsin(bt)$, where $A = 3$ and $b = 2\pi$ were chosen to be similar to the average trajectories of the probing movements that the participants of Experiment 1 performed, and gs was chosen to be one of the three tactor displacement gains (33, 66, and 100 mm/m) that we used in this paper. Each tactor displacement gain was repeated 10 times throughout the experiment in random order.

Appendix 1—figure 4 presents the results of the passive control experiment – the average grip force trajectories for the different levels of tactor displacement gain. The trajectories were analyzed in the same manner as those of Reactive Experiment 2, that is, they were time-normalized and aligned so that 0 was the onset of the tactor movement, and one was the end. Similarly to the results of Control Experiment 1, initiation of the stretch led first to a decrease, and then to an increase, in the grip force trajectory, and the magnitude of both the decrease and the increase was dependent on the tactor displacement gain. The pattern of the trajectory is similar to the pattern in the Reactive Experiment 2, but the magnitude of the artifact is smaller by a factor of 8.

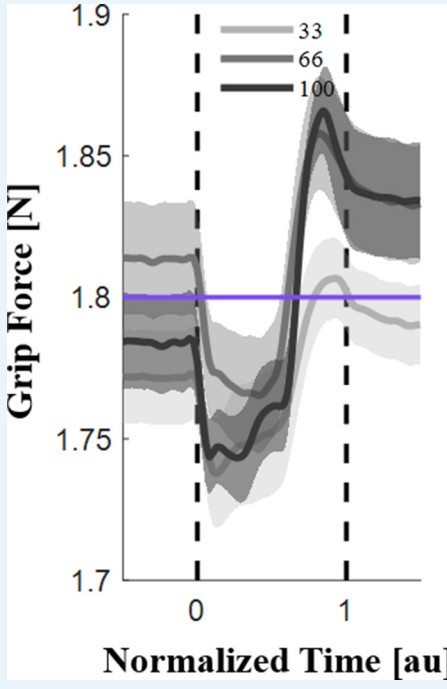

**Appendix 1—figure 4.** Control experiment 2 (passive experiment). Average grip force artifact trajectories for the different levels of tactor displacement gain. Shading represents the standard errors, and the black vertical dashed lines represent the time of the interaction with the stretch. The purple line represents the grip force level required for the initiation of the stretch.

## Discussion

The reactive grip force pattern was very surprising. At the beginning of an interaction, a tactile stimulus is similar to any mechanical perturbation. It is well-established in the literature that during unexpected load force perturbations, the grip force is automatically adjusted (*Cole and Abbs, 1988*; *Johansson and Cole, 1992*; *Cole and Johansson, 1993*). Recordings from tactile afferents during trapezoidal load force perturbations to the digit with different rates of loading (*Macefield et al., 1996*) revealed that the mean firing rate was scaled by the slope of the loading ramp. This prompted our initial prediction that the reactive response would be an increase in the grip force. Furthermore, we posited that the size of this effect would increase linearly with increasing tactor displacement gain, and would appear more than 90 ms after stimulation (*Dimitriou et al., 2012*; *Pruszynski et al., 2016*; *Crevecoeur et al., 2016*; *Crevecoeur et al., 2017*). Instead, in response to the artificial stretch, participants first strongly decreased their grip force and only then increased it, resulting in an overall moderate increase that depended linearly on the tactor displacement gain. The peak of the decrease was roughly in the middle of the probe, close to the peak of the stretch and the load force, and about 150 ms from the onset of the stretch. The peak of the increase was after the stretch, and the load force returned to zero about 300 ms from the onset of the stretch.

In both reactive experiments, the magnitude of the initial decrease and the subsequent increase of the grip force was greater with larger tactor displacement gain. In addition, the results of Reactive Experiment 2 indicates that the response is more pronounced when the baseline grip force is larger. This can explain the increase in the differences between the peak decrease and peak increase in the second and seventh reactive responses in Reactive Experiment 1. The predictive grip force increased between the second and seventh probing movements. Therefore, with a larger underlying grip force, the magnitude of the reactive response is also expected to increase.

The trajectory of the reactive grip force control was similar between active interaction with the elastic force field (Reactive Experiment 1), and maintenance of constant grip force (Reactive Experiment 2). In the former, participants could anticipate the stretch based on their planned movement trajectories, especially during the seventh probes. Conversely, in the latter, the trajectory of the tactor displacement was surprising. Nevertheless, while it is impossible to compare these responses directly due to different units, the trajectories are qualitatively similar.

We do not have a conclusive explanation as to why the reaction to skin-stretch would be an initial release of grip. One possibility is that the release was the result of an unpleasant or painful interaction, however none of our participants indicated any discomfort, nor did they report any when asked about their experience at the end of the experiment. Another possibility is the attempt to reduce the contact impedance between the fingers and the device due to possible contact instability (*White et al., 2018*).

It is important to note that the reactive grip force response we report here may be a specific response to our skin-stretch device, and not a general response to any artificial tactile stimulation. In addition, these results should be considered with caution because of the resemblance between the reactive grip force trajectories and the trajectories of the artifact of tactor movement, which we characterized in the two control experiments. However, the participants' reactive response was much greater than the artifact, suggesting that the artifact is not a plausible explanation for the entire response. Future studies, using other designs of tactile stimulation devices, are needed to pinpoint the origin of this surprising reactive pattern.

## Appendix 2

### The mapping between the actual grip force and the downscaled measurement of the grip force

Due to the placement of the force sensor that we embedded in our skin-stretch device, a downscaled version of the grip force was measured. We were therefore interested in performing a validation experiment to characterize the relationship between the actual grip force and the downscaled version that we measured. To do this, we connected an additional force sensor (ATI, Nano17) to the outer part of the aperture. This force sensor allowed us to measure the exact grip force applied between the two fingers. We then compared between the force measured by the two force sensors to characterize the mapping between the exact and downscaled grip force.

To connect the additional force sensor to the outer part of the device, we 3D-printed two connectors and attached them to both sides of the external force sensor (*Appendix 2—figure 1*). Connector one rigidly attached the force sensor to the aperture of the skin-stretch device, which is where the participants gripped the device during the experiments. Connector two was used to prevent direct contact between the finger and the sensor to eliminate the effect of skin temperature on the output of the force sensor. The connection of the external force sensor created a bias of 1N to the downscaled measurement of the force sensor that was embedded in the skin-stretch device, and therefore, we subtracted this bias from the recorded grip force.

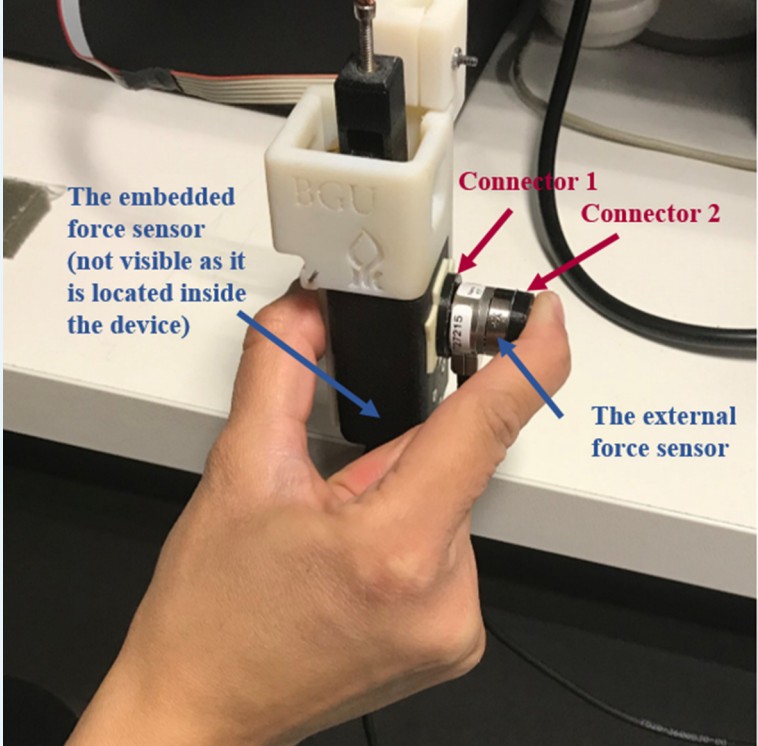

**Appendix 2—figure 1.** The experimental system of the grip force mapping experiment. The system consists of two force sensors: (1) The embedded force sensor, which was mounted on the lower end of the device so that participants did not place their fingers directly above the force sensor (not visible here because it located inside the outer black part of the aperture. (2) An external force sensor, which was connected to the outer black part of the aperture, where the participants gripped the device during the previous experiments.

The two force sensors were attached to the computer using the same PC DAQ (NI-6323) and recorded the applied forces simultaneously. We disconnected the skin-stretch device from the haptic device, and asked each of five participants to exert normal force on the device five times (without tactor displacement movement). We asked five different users to participate in this calibration experiment because the dimensions of the finger and other factors influence the mapping between the actual applied forces and the forces that are measured with our embedded force sensor. The recorded grip forces in this control experiment were in the range observed in our experiments.

*Appendix 2—figure 2a* presents the grip force recorded by each of the two force sensors as a function of time of a typical participant, and *Appendix 2—figure 2b* depicts all the datapoints of all the participants (color-coded). For each participant, we fitted a regression line between the exact and the downscaled grip force and examined the goodness of fit ($R^2$). *Appendix 2—figure 2b* presents the linear regression of all the participants with $R^2 = 0.929, 0.907, 0.769, 0.937, 0.984$.

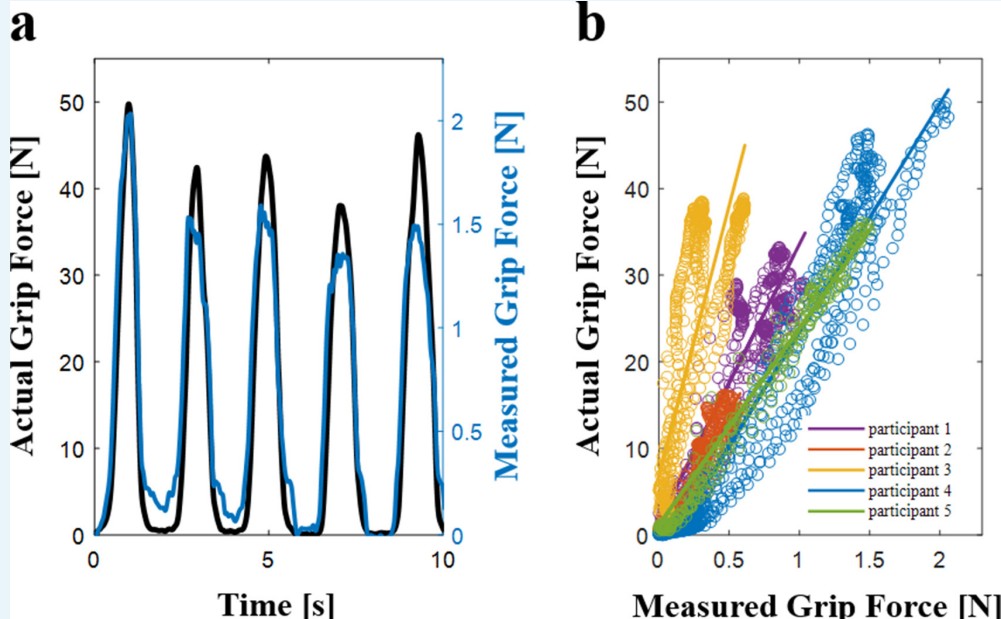

**Appendix 2—figure 2.** Control Experiment. (**a**) The actual and the measured grip force as a function of time of a typical participant. (**b**) The linear regression between the actual and the measured grip force of all the participants (N = 5). The abscissa is the downscaled measured grip force using the embedded force sensor and the ordinate is the actual grip force recorded using the external force sensor. The markers represent the recorded grip force and the solid lines show the fitted regression lines.

Aalthough all the participants demonstrate a near linear relationship between the measurement of the actual grip force and the downscaled measurement, the fitted models differ slightly amongst them (that is, different intercept and slope). The difference stems from the fact that the grip force measurement with the embedded force sensor depends on the size of the participants' fingers (due to division of pressure between the tactor and the aperture) and on the way they hold the skin-stretch device. This is also true in all the experiments we reported throughout the paper, and therefore even though this experiment allows for calculating a mapping between actual and measured grip forces, we decided to report the measured downscaled forces rather than an estimate of the actual forces. Nevertheless, our experimental design is entirely within-participants, and therefore, the interpretation of all the trends that we observed in this study is not affected by this choice.

