## [Decision Letter]

**Acceptance summary:**

The revisions have really helped clarify the significance of the paper. The study examines the control of the grip force (for example picking a cup of coffee), an action that requires accurate modulation of the grip to ensure the force is appropriate and well-timed to prevent the cup from slipping, and at the same time manipulate the cup in space. This problem is quite general for sensorimotor control. The major challenge is not just to control the immediate action, but also to anticipate the future.

**Decision letter after peer review:**

Thank you for submitting your work entitled "Stretching the skin of the fingertip creates a perceptual and motor illusion of touching a harder spring" for consideration by eLife. Your article has been reviewed by three peer reviewers, including Eilon Vaadia as the Reviewing Editor and Reviewer #1, and the evaluation has been overseen by a Reviewing Editor and a Senior Editor.

Three reviewers read the manuscript. All three agree that the experiments address a fundamental question. On the positive side, we conclude that the researchers use a neat system to investigate the effect of stretching the skin of the fingertip while performing movements in a force field. The force field simulated the stiffness of a spring in the virtual environment. The goal was to determine how the tactile and kinesthetic feedbacks are combined and influence both stiffness perception and grip force control. They monitored the actions of the subjects as well as the reported perception of stiffness in different conditions and two versions of experiments.

As a whole, this seems to be a detailed set of experiments. Answering the research questions may be quite important and significant as a demonstration of the notion that perception and sensorimotor learning do not evolve merely by some higher level of sensory processing, but by an active and adaptive process of integrative nature, involving estimation of the consequences of actions and adaptation of sensorimotor internal model.

However, the only truly innovative part comes from Experiment 2, while the results of Experiment 1 are not new (credit is given to the PI of the current paper for the previous publication). The results of Experiment 2 include a robust and original data showing a negative reactive response of Grip Force to skin stretching, but Experiment 2 requires a more serious treatment (see comments below).

In general, the three reviewers had several major concerns. Some of these concerns could possibly be addressed by extensive revisions. But we regret that some more critical concerns, raised by two of the reviewers led to our decision to reject the paper.

While we agree that the topic is interesting, we estimate that the necessary revisions are far too extensive to allow in a reasonable time frame a resubmission of a revised manuscript. We therefore reject your manuscript in the present form. If you decide to conduct additional control experiments, add and clarify some analyses, edit the figures, clarify interpretations, and re-write the paper using a more explicable and comprehensive approach, which facilitates understanding the message, on one hand, and help the general reader following the technical details on the other, we would consider evaluating such an improved manuscript as an entirely new submission.

Essential revisions:

1) Perhaps the most interesting aspect of the study is the impact of skin stretch on perception and action. But this is not new (Experiment 1). We have known for some time that skin conformance biases tactile judgement (Vega Bermudez and Johnson, 2004). The group has studied the effects of the skin stretch device on the perception of stiffness and motor control, as previously published in IEEE (e.g. Quek et al., 2014). From my understanding of the manuscript, the current contribution to this interesting phenomenon is only incremental with regards to such previous work published by the group. This concern does not address research question of Experiment 2, but unfortunately, the results of the experimenters are not fully understood and require some more control experiments (see below).

2) While the experiments, especially Experiment 2 are in some way elegant, they did neither lead to conclusive and reliable answers nor a comprehensive notion of the neuronal process that explains the so-called "illusion". As listed below, some more analyses and control experiments are needed to reach a clear a convincing conclusion.

3) The three reviewers complain that the paper is quite hard to read. We believe that more efforts are necessary to communicate the results and interpretations to the general reader who is less of an expert in the details of the specific technology and methodologies used here. A substantial revision of the manuscript structure and style is required. Some technical issues (like the artefact problem) could be moved to supplementary materials. Some of the analyses (like the regression to find the intercept in Experiment 2) could be better explained to help the reader follow the rationale and convince that the goal of Experiment 2 is indeed achieved reliably.

In general, the manuscript suffers unclear and complex wording; the introduction is inflated, providing many dissociated ideas and concepts without a clear narrative. The results section is overly long and detailed, while some analyses are missing and some could be moved to supplementary section (for the benefit of experts).

4) One of the reviewers felt that the impact of the study in designing prosthetic limbs and surgical devices is overstated, as the current study doesn't provide any new tools or opportunities for these developments with regards to the state of the art. If the authors wish to promote the usage of skin stretching to improve e.g. control of prosthetic limbs, they should study that.

5) The authors are advised to better describe and justify their statistical tests and presentation of the data. See some derails below.

6) Why did the authors choose ANCOVA (and not ANOVA), what do the degrees of freedom represent? How exactly was the interaction of interest constructed?

7) Description of Experiment 1 in the results section is unclear: "In some of the force fields, they also received skin- stretch feedback via a custom-build tactile stimulation device" In which of the force fields?

8) The labelling of the axes is sometimes unclear e.g. Fig 2A - the 'probability of responding comparison' is not well defined.

9) Authors claim that increasing the skin-stretch gain causes more grip force peaks to appear. This is not evident from the presented figure (Figure 3). How was it quantified? By visual inspection?

10) The explanation of how the reactive component of grip force control was isolated in Experiment 2 should be made more clearly.

11) Why do we only see the phase differences between grip force and load force in Experiment 1 (Figure 3), but not in Experiment 2 (Figure 5)? Similarly, why is the 'irregular pattern of decrease accompanied by increase' only apparent in Experiment 2? Weren't those experiments performed in the exact same way, except for the controlled number of probing movements and presence of the stretch-catch trials?

12) The quantitative analysis of the peak grip force to peak load force ratio is very problematic in this paper because the tactile stimulation generates major time shifts between Grip Force and Load Force (and multiple peaks for the Grip Force). Therefore, the interpretation of this ratio is very hard at this stage. The reported effects could be due to the predictive feedforward component or to the reactive feedback component or to their interaction. This is why the panel b of Figure 6, for instance, is probably misleading, because there is an asynchrony between Grip Force and Load force in the data (see also comments below). The authors overlook this problem of asynchrony in the paper (see Discussion section: this major issue of asynchrony and multiple peaks in the Grip Force response is ignored).

13) The last panel of Figure 5 is a very important result from this study: the 7th stretch-catch probe shows that the feedforward predictive component of the Grip Force (in anticipation of a skin stretch that does not occur) is characterized by a single peak (and is larger than if no skin stretch is expected). It also shows that the reactive tactile feedback response reduces the Grip Force response (see double peaks in other trials, same panel). This reduction of the Grip Force was not expected and the authors should better convince the reader that it is not an artefact of their stimulus. I suggest for instance to test the response to the skin stretching when maintaining a STABLE Grip Force, at different typical values, within the Grip Force range tested in the study. Is the Grip Force response also a reduction in this case?

14) The authors mention in the discussion section that: "The reactive grip force pattern was very surprising". Indeed, they expected an increase in Grip Force instead of a decrease in reaction to the tactile stimulus. The authors should consider the possibility that the tactile reactive feedback (reduction of Grip Force) might be due to a general mechanism that could be some kind of response to a nociceptive input when there is a mismatch between what you expect from prediction and what you feel at your fingertips (think about a bug that you hold between your fingers and that tries to escape). This response could be specific to the situation when the Grip Force varies (active movement) and not during static holding (see control experiment suggested above) or could alternatively be present in both conditions.

15) The authors also mention in the discussion section: "This prompted our initial prediction that the reactive response would be an increase in grip force 50-90 msec after the stimulation". This prediction is not correct on the basis of existing literature (Dimitriou et al., 2012; Pruszynski et al., 2016, Crevecoeur et al., 2017). Indeed, the measured EMG response to a tactile stimulus is around 90ms. Therefore, a change in Grip Force is not expected before 120 to 150ms, which is compatible with the reported data for the peak of decrease: "The peak of the decrease was roughly in the middle of the probe, close to the peak of the stretch and the load force, and about 150 msec from the onset of the stretch.".

16) What is puzzling in the data is that the subjects are exposed to a repetition of tactile stimuli but lack adapting their motor response efficiently. Indeed, they cannot inhibit the reduction in Grip Force due to the tactile feedback and consequently show a local minimum of Grip Force when the Load Force is maximum. This is clearly a very bad strategy that is repeated by the subjects who fail to adapt in an efficient way (they react to the increased uncertainty due to the tactile stimulus by boosting the feedforward component). It can only be partially compensated by increasing the feedforward predictive Grip Force component (to compensate for the expected and inefficient reduction of Grip Force after the tactile stimulus) (subsection “Skin-stretch increased the feedforward component of grip force”).

17) In Experiment 1, the subjects performed several probing movements for each stimulus. When there was a tactile stimulus, they adapted their strategy across repetitions (see Figure 3 last panel and Figure 4B for a comparison between first and last ratio). Therefore, it is not clear what induced the change in PSE. Is it the tactile stimulus or is it the build-up of increased feedforward Grip Force across repetitions with the tactile stimulus? A control experiment involving a test with a single probing movement is necessary from my point of view. It would allow disambiguating the role of the tactile stimulus versus feedforward component in assessing stiffness. An alternative idea would be to test perception of stiffness specifically during catch trials (it remains to be determined how this could be done in a protocol). It would also allow separating the feedforward component (that is increased in catch trials) from the feedback reactive response to the tactile stimulus.

18) In the discussion section the authors mention that: "another possible explanation for the increased grip force is that the motor system of the participants could have interpreted the additional tactile stimuli as indication of a more slippery contact surface, which would have thus been entirely unrelated to the perceptual illusion of a harder spring." This is a very interesting idea. I encourage the authors to add analyses related to the correlation between the Grip Force and Load Force (phase diagram) during the first open-loop phase of the probing movements (Johansson and Westling, 1984). The correlation between the derivative of GF and LF should also be very informative. This would allow assessing the feedforward predictive component of Grip Force even during standard probing movements (Experiment 1) and not only during catch trials (Experiment 2).

19) In the discussion section, the authors mention that: "Future studies, probably with other designs of tactile stimulation devices, are needed to pinpoint the origin of this surprising reactive pattern." Given that the same group already reported the finding of the change in the perception of stiffness following a tactile stimulus (papers by Quek et al.), I wonder whether better characterizing and understanding the nature of the reactive pattern is not necessary to make the present study truly new, interesting and innovating.

[Editors' note: further revisions were suggested prior to acceptance, as described below.]

Thank you for resubmitting your work entitled "Stretching the skin immediately increases perceived stiffness and slowly increases the predictive control of grip force" for further consideration by eLife. Your revised article has been evaluated by Richard Ivry (Senior Editor) and two reviewers, including Eilon Vaadia (the Reviewing Editor).

Summary:

The authors address a specific question, related to control of the grip force that all of us use with no difficulty when we grasp an object like a paper-cup of coffee. This action requires accurate modulation of the grip, at an accurate and well-timed force, to prevent the cup slippage, (spilling the coffee), and at the same time avoiding crushing the paper cup. Investigation of this question is related to many aspects of action-perception behaviors and beyond. The major challenge in all cases is not just the action itself but in the prediction of the future, given the present (and past) concepts and actions. Therefore, the potential projection of this study to more general science fields is part of the general significance of this study.

Here, the authors investigated the effect of stretching the skin of the fingertip while performing movements in a force field. The force field simulated the stiffness of a spring in a virtual environment. The goal was to determine how the tactile and kinesthetic feedbacks are combined and influence both stiffness perception and grip force control. The authors also used catch trials to try and separate the three components of the Grip Force response: predictive feedforward baseline, predictive feedforward modulation and reactive feedback. The authors address an interesting and very important question related to the role of tactile inputs on perception and motor control. Experiment 2 based on stretch-catch probes reports very robust and original data showing a negative reactive response of Grip Force to skin stretching. The authors also performed several control experiments to better assess the limitations of their finger stimulation device. The results on the influence of tactile stimulation on the perception of stiffness are very robust and confirm earlier reports from the same group. The demonstration that higher levels of artificial skin-stretch increased the predictive grip force modulation in anticipation of the load force and the finding that the increase developed throughout each trial are important. Thus, it is also important to improve and quantify the temporal aspects of the measurements.

The manuscript has been greatly improved but there are some remaining issues that need to be addressed before acceptance.

1) We ask that you provide a better characterization of the evolution of grip force over time.

2) The results on the influence of tactile stimulation on the perception of stiffness are very robust and confirm earlier reports from the same group. In contrast, the results on the influence of tactile stimulation on Grip Force control remain very difficult to interpret despite the additional control experiments performed by the authors. This could be partly improved by conducting additional analyses as suggested below.

a) In their analyses, the authors try to separate the baseline and modulation of Grip Force (Results section): "To separate between the effect on each of these two components, we performed a linear regression analysis in the grip force-load force plane." This is illustrated in Figure 4 with phase plots of Grip Force versus Load Force. However, this figure shows that the relationship between Grip Force and Load Force is not very linear. This is also clear in Figure 2 because the Load Force profile is very stereotyped and short in time in comparison with the Grip Force profile that starts increasing well before the increase in Load Force in anticipation of the contact with the virtual object. This observation is classical in all studies investigating collisions between (virtual) objects. In that case, it is much more appropriate to measure the baseline Grip Force as the average Grip Force level between two consecutive trials (when Grip Force does not vary much over time, see Figure 2D). It is also more appropriate to measure the modulation when both Grip Force and Load Force vary quickly during the early part of the contact. This kind of analysis was recently used in the recent paper of Kuling et al., (2019). In the same paper, these authors also propose a method to compute an estimate of the predictive component of the feedforward Grip Force on the basis of a regression analysis taking into account Grip Force at contact (with the virtual object) and the derivative of Grip Force at contact. This method was validated on catch trials when there was no contact with a virtual object. This method is interesting in the context of the present study because it would allow further investigating the gradual build-up of Grip Force feedforward component across the repetition of trials with tactor displacement. The following reference could also be cited (Grip Force during collisions in a virtual reality setup): White et al., (2011).

b) In the Materials and methods section and throughout the paper, the authors acknowledge that their "device only measures a downscaled version of the grip force". Of course, this introduces a major limitation in the interpretation of the experimental data related to the measure of Grip Force. At this stage, there is no quantification of the relationship between actual Grip Force and the measured downscaled version of Grip Force. We do not even know if the relationship will be linear or non-linear given the complex mechanics of the object / finger pad interaction (pressure is not constant in the contact area). In their paper, the authors tried to address this important issue but in a way that is not satisfactory given the importance of Grip Force measurement. I strongly recommend performing the following very simple control experiment to clarify this issue. The authors should simply disconnect their manipulandum device from the haptic robot and put it flat on an extra (1D) force sensor. They should then instruct a few human subjects to exert a pure normal (vertical) force on the device over the range of recorded Grip Force in the experiments. They should then compare the Grip Force measure with the device with the true total normal force measured by the additional force sensor to assess the "downscaling" of the measured Grip Force with the device. This could be done with or without tactor displacement for a more thorough control measurement.

3) The experiment does not show directly any "internal representations" of the well-demonstrated illusion. In general, the term It is an over-used, and means very little. Of course, the authors can see it their way, and I do not demand the elimination of terms. I counted 28 times (twice in the Abstract!) I do recommend to reduce this number or abandoning this term, leaving the issue of neuronal basis to speculation/interpretation alone. For example, the text of the authors:

"This suggests that the perceptual illusion is a result of a high-level cognitive process, independent of the gradual development of the internal representation that causes an increase in the predictive grip force control.

Could be simpler... like:

"This suggests that the perceptual illusion is a result of an adaptive predictive process, independent of the gradual development of grip force control during the action".

4) The results on the influence of tactile stimulation on the perception of stiffness are very robust and confirm earlier reports from the same group. In contrast, the results on the influence of tactile stimulation on Grip Force control remain very difficult to interpret despite the additional control experiments performed by the authors. This could be partly improved by conducting additional analyses as suggested below.

5) The authors report (Results section) that "participants applied more grip force per amount of kinesthetic load force when interacting with force fields with higher levels of tactor displacement gain." This observation makes sense because the skin stretch induces a large decrease of Grip Force around the peak of Load Force. It is particularly clear for large tactor displacement (100 mm/m, Figure 2D) where there is a clear dip in Grip Force around the peak of Load Force (see comparison between trial 6 and 7 in Figure 2D: difference in Grip Force of 2 N around the peak Load Force). Therefore, this dip in Grip Force around the peak of Load Force increases the risk of slippage which is compensated by an increase of baseline and feedforward Grip Force components (see comments below).

6) Very interestingly, the authors mention in the same paragraph (Results section) that "This trend is significantly higher in the seventh probing compared to the second". This is a very important observation because it shows that the increase in baseline and feedforward Grip Force components is clearly decoupled from the perception of increased stiffness that is shown in the manuscript to take place after a single trial as demonstrated in Experiment 2 and Figure 5 (subsection “Skin-stretch caused perceptual overestimation of stiffness”): "We can therefore conclude that the perceptual illusion was formed immediately, after a single probing movement into the force field, and before the internal representation that we identified in Experiment 1 was formed. "

7) Results section: "This suggests that following repeated exposure to the artificial skin-stretch stimulation, the participants either increased their internal representation of the expected force (likely due to increase in the representation of the stiffness of the field) or of the slipperiness of the device." Again, see comments above: the fact that the perception of stiffness occurs much faster and that the measured effects on catch trials 2 and 7 (Figure 4F) are very different seems to demonstrate that the reported effect is not due to gradual increase in the internal representation of the expected force but more likely to a slower adaptation of Grip Force potentially due to an increased risk of slippage ("reactive" dip in Grip Force).

8) One key experiment for the interpretation of the Grip Force data of the current study is the Reactive Experiment 2 reported in the Supplementary File. I believe that it is crucial to fully exploit this data set to better understand the effect of tactor displacement on the "reflex" Grip Force response reported in the main experiment. I even think that a better controlled experiment based on the analyses of this Reactive Experiment should be published as an independent study before publishing the current set of experiments that cannot be satisfactorily interpreted in terms of Grip Force control at this stage. I am convinced that it could be a major contribution to our understanding of the very important question related to the role of proprioceptive versus tactile feedbacks in motor control (Crevecoeur et al., 2017). The device used by the authors allows investigating the impact of a pure tactile stimulation on the motor response (Grip Force reaction). I recommend redoing this reactive experiment and recording EMG signals together with the measurement of the Grip Force reaction. This will allow measuring the delay of the motor response and better characterizing the timing of the reaction which is crucial to understand the results of the current study (the delay seems very short in Figure 10: shorter than 20ms?).

9) Figure 7 shows the ratio between peak Grip Force and peak Load Force. It is not clear what this analysis means because in Figure 6 the peaks of Grip Force and Load Force are not synchronized! In the current version of the manuscript (and reported analyses) the interpretation of Grip Force data is very weak (see also Materials and methods section).

10) In the Discussion section, the authors mention that "in our case, increasing the grip force reached the perceptual illusion over repeated interactions". I am not sure that the interpretation of the authors is correct (see also comments above) because there is an illusion in terms of stiffness but not in terms of tactile sensation: there is an actual stretch of the skin, a "reflex" dip in Grip Force, and therefore a sensation of partial slip or strain in the fingertip. Moreover, the time course of the two phenomena is very different: immediate illusion in comparison with progressive build-up of Grip Force anticipation, pointing to very likely different mechanisms.

---

## [Author Response]

As a whole, this seems to be a detailed set of experiments. Answering the research questions may be quite important and significant as a demonstration of the notion that perception and sensorimotor learning do not evolve merely by some higher level of sensory processing, but by an active and adaptive process of integrative nature, involving estimation of the consequences of actions and adaptation of sensorimotor internal model.However, the only truly innovative part comes from Experiment 2, while the results of Experiment 1 are not new (credit is given to the PI of the current paper for the previous publication). The results of Experiment 2 include a robust and original data showing a negative reactive response of Grip Force to skin stretching, but Experiment 2 requires a more serious treatment (see comments below).In general, the three reviewers had several major concerns. Some of these concerns could possibly be addressed by extensive revisions. But we regret that some more critical concerns, raised by two of the reviewers led to our decision to reject the paper.While we agree that the topic is interesting, we estimate that the necessary revisions are far too extensive to allow in a reasonable time frame a resubmission of a revised manuscript. We therefore reject your manuscript in the present form. If you decide to conduct additional control experiments, add and clarify some analyses, edit the figures, clarify interpretations, and re-write the paper using a more explicable and comprehensive approach, which facilitates understanding the message, on one hand, and help the general reader following the technical details on the other, we would consider evaluating such an improved manuscript as an entirely new submission.

Thank you for encouraging us to revise and resubmit our paper as a new submission. We carefully analyzed the suggestions, added new experiments (including control experiments), added and clarified the analyses, completely restructured the paper such that the message and the contribution are clearer. Throughout this reply we will refer to Experiment 1 from the original manuscript as original Experiment 1, and the same for our second experiment from the original manuscript. These experiments differ from those presented in the new manuscript in several ways. First, we switched the order between them to lead to a clearer and more coherent paper. Furthermore, we added an additional group to one of the experiments. To focus on the main message of the paper, we moved all the analyses of the reactive component of the grip to the Supplementary file, together with the original control experiment and two new experiments. We improved the figures and sent the paper to a thorough editing by a professional English editor who helped to rewrite the paper in a more coherent and flowing manner and sharpen our message.

Essential revisions:1) Perhaps the most interesting aspect of the study is the impact of skin stretch on perception and action. But this is not new (Experiment 1). We have known for some time that skin conformance biases tactile judgement (Vega Bermudez and Johnson, 2004). The group has studied the effects of the skin stretch device on the perception of stiffness and motor control, as previously published in IEEE (e.g. Quek et al., 2014). From my understanding of the manuscript, the current contribution to this interesting phenomenon is only incremental with regards to such previous work published by the group. This concern does not address research question of Experiment 2, but unfortunately, the results of the experimenters are not fully understood and require some more control experiments (see below).

Thank you for highlighting the lack of clarity in the contribution of this work in comparison with prior studies of the effect of skin-stretch on perception and action. In the original submission, we failed to highlight that while the perceptual augmentation was well documented in (Quek et al., 2014), the effect of artificial skin-stretch on grip force was only addressed at the level of an increase in the average grip force (Quek et al., 2015a, 2015b), whereas the modulation of grip force with load force and stretch was not studied. Our original Experiment 1 addressed this modulation in a setting in which participants used free exploration to assess the stiffness of the force fields. Meaning, participants were free to probe the force field as many times as they wished. The results of this experiment led to the design of the original Experiment 2, which addressed this question in further detail by investigating the effect of the skin-stretch on each of the components of grip force control separately. Based on these analyses, we uncovered the effect of skin-stretch on the internal representation that is used in grip force control. In this new submission, we reorganized the paper, such that this novel result is presented first. Additionally, we added a new group to the original Experiment 1 such that we were able characterize the time scale of the formation of the perceptual effects of artificial skin-stretch during subsequent probing.

The main findings of the new revised submission are as follows:

1) High levels of artificial skin-stretch increased the predictive grip force modulation in anticipation of the load force. The increase developed throughout each trial with repeated probing interactions with the same elastic force field.

2) Artificial skin-stretch caused participants to not reduce their safety margin, even after an internal representation was formed. This result is in contrast to the well-known (and reproduced in our study) result that, without artificial stretch of the skin of the fingertips, during repeated interactions with the same elastic force field, the safety margin is reduced.

3) The perceptual augmentation of stiffness (the illusion of touching a harder spring) is formed immediately upon the initial contact with the object, and does not depend on the build-up of the internal representation that is used for the predictive control of grip force.

4) The reaction to stretch of the skin of the fingertips is a release of the grip. However, in spite of running several control experiments, we are still unsure if this response is specific to our device or a general response to artificial stretch. As such, this result leaves more open questions than answers, and we have moved the entire analysis and discussion about this result to the Supplementary file.

We completely reorganized the paper and added clarifications of the contribution to the Abstract, Introduction, and Discussion section.

2) While the experiments, especially experiment-2 are in some way elegant, they did neither lead to conclusive and reliable answers nor a comprehensive notion of the neuronal process that explains the so-called "illusion". As listed below, some more analyses and control experiments are needed to reach a clear a convincing conclusion.

We appreciate this point, and agree that our behavioral approach cannot pinpoint the neuronal processes that explain the perceptual illusion. However, the new perceptual experiment (new Experiment 2, Group 1) sheds light on the forming of the illusion. We found that it is built almost entirely after a single probing movement into the elastic force field, and does not depend on the slower formation of the internal representation that is reflected in the predictive grip force control. These findings lead to important questions for future work on identifying the specific neural pathways that allow for such fast formation of the perceptual illusion that is, at least partially, dissociated from the slower effect on grip force. We clarified this point in the revised discussion section:

“Consistently with previous studies (Vega-Bermudez and Johnson, 2004; Quek et al., 2013, 2014; Park, Oh and Tan, 2018), we found that adding artificial tactile feedback augmented the perceived stiffness, and that the augmentation was a linear function of the tactor displacement gain. Our findings show that this perceptual illusion is immediate, and does not depend on repeated interactions with the force field. This suggests that the perceptual illusion is a result of a high-level cognitive process, independent of the gradual development of the internal representation that causes an increase in the predictive grip force control. This is consistent with recent findings regarding the fast timescales of explicit cognitive strategies in motor adaptation, compared to the slower implicit adaptation mechanisms (McDougle, Bond and Taylor, 2015). In fact, the immediate perceptual effect observed in Experiment 2 was evident after the participants were exposed to exactly the same amount of information as the participants in Experiment 1 had when they planned their control of grip force in the second stretch catch probe. In the second stretch catch probe, only the baseline grip force component, which reflects uncertainty, increased due to the skin-stretch, and the modulation, which reflects the internal representation, was independent of the amount of stretch. Conversely, the perceptual effect after one probe was similar to the perceptual effect after free exploration, and was linearly related to the amount of stretch."

We also added the requested control experiments (see response to comment 12).

3) The three reviewers complain that the paper is quite hard to read. We believe that more efforts are necessary to communicate the results and interpretations to the general reader who is less of an expert in the details of the specific technology and methodologies used here. A substantial revision of the manuscript structure and style is required. Some technical issues (like the artefact problem) could be moved to supplementary materials. Some of the analyses (like the regression to find the intercept in Experiment 2) could be better explained to help the reader follow the rationale and convince that the goal of Experiment 2 is indeed achieved reliably.

We clarified the rationale behind this regression analysis in the Introduction and the results section:

“Feedforward control is used to adjust our grip force in accordance with the expected slipperiness and weight (or load force) of the object (Flanagan and Wing, 1990, 1997; Flanagan et al., 2003; Danion and Sarlegna, 2007; Danion, Descoins and Bootsma, 2009; Leib, Karniel and Nisky, 2015; Leib, Rubin and Nisky, 2018; White et al., 2018). Baseline grip force is maintained to create a safety margin, and is increased when one experiences uncertainty regarding the load force (Gibo, Bastian and Okamura, 2013; Hadjiosif and Smith, 2015). Predictive modulation is calculated based on an internal representation of the object dynamics, which is learned by the nervous system (Johansson and Cole, 1992; Shadmehr and Mussa-Ivaldi, 1994; Kawato, 1999; Davidson and Wolpert, 2004) and is updated during repeated interactions (Donchin, Francis and Shadmehr, 2003).”

“The intercept of this regression represents the baseline grip force; i.e. the amount of grip force that was applied by the participants when no external load force was applied by the haptic device. The slope of the regression represents the modulation of the grip force that anticipates the load force. In typical manipulation tasks of grasping and lifting, the slope is linked to the “slip ratio” of an object (Flanagan and Wing, 1990), and when interacting with elastic objects, to a combined effect of the slip ratio and the internal representation of the load force (Flanagan and Wing, 1990; Leib et al., 2018, 2015).

In addition, we highlighted the calculation of the slope and the intercept of the grip force load force regression in the Materials and methods section:

“To separate the modulation of grip force in anticipation of the load force from the baseline grip force control, we analyzed the grip force-load force regression, similar to the analysis in (Flanagan and Wing, 1990; Leib et al., 2015; Nowak et al., 2002). We fit a two degrees-offreedom regression line (slope and intercept) to the trajectory in the grip force-load force plane for each of the stretch catch probes (second and seventh probing movements), as well as for trials with a tactor displacement gain of 0 mm/m. The intercept represents the baseline grip force; i.e. the amount of grip force that participants applied when no external load force was applied by the haptic device. The slope of the regression represents the modulation of the grip force in anticipation of the load force.”

In general, the manuscript suffers unclear and complex wording; the introduction is inflated, providing many dissociated ideas and concepts without a clear narrative. The results section is overly long and detailed, while some analyses are missing and some could be moved to supplementary section (for the benefit of experts).

We heavily edited and shortened the manuscript to address these issues. We shortened the Introduction and the Discussion section to focus on our message, which is identifying the contribution of tactile stimulation of the fingertips during interaction with elastic objects on the internal representations of stiffness that are used for perception and the predictive control of grip force. We moved some of the original, and new, control experiments to the Supplementary file, as well as the analysis of the reactive component of grip force. We also sent the manuscript to scientific editing to improve the wording, flow, and structure of the paper.

4) One of the reviewers felt that the impact of the study in designing prosthetic limbs and surgical devices is overstated, as the current study doesn't provide any new tools or opportunities for these developments with regards to the state of the art. If the authors wish to promote the usage of skin stretching to improve e.g. control of prosthetic limbs, they should study that.

We removed the paragraph describing these future impact applications.

5) The authors are advised to better describe and justify their statistical tests and presentation of the data. See some derails below.

We revised our description of the statistical analyses – see details in the response to the individual comments.

6) Why did the authors choose ANCOVA (and not ANOVA), what do the degrees of freedom represent? How exactly was the interaction of interest constructed?

We apologize for using potentially confusing terminology. The analyses that were originally performed were generalized linear model (with continuous, categorical, and random factors) or regression with repeated measures (with continuous and random factors). We added a clarification in the subsection “Data analysis”:

“In the following analyses of grip force control, for each of the dependent variables separately, we used a generalized linear model with a continuous factor (tactor displacement gain, df=1), a categorical factor (initial and late probing, df=1), a random factor (participants, df=N-1, where N is the number of the participants in each experimental group), and their respective interactions. In the analysis of the perception (separately for PSE and JND), we used repeated measures regression with the tactor displacement gain as the independent variable (df=1), participants as a random categorical factor (df=N-1), and their interaction”.

We revised the subsection “Data Analysis” to clarify these terms in each of the statistical models separately. For example:

“We fit a repeated-measures General Linear Model to each the slope and the intercept separately as the dependent variable in two separate statistical analyses, using the anovan MATLAB function. The independent variables were the tactor displacement gain (continuous, df = 1), the probing movement (categorical, 2 levels - second or seventh, df = 1), and participants (random, df = 9). The model also included interactions between independent variables.”

7) Description of Experiment 1 in the results section is unclear: "In some of the force fields, they also received skin- stretch feedback via a custom-build tactile stimulation device" In which of the force fields?

We revised the text to clarify this point:

“In each trial, participants interacted with two different force fields, designated standard and comparison, and decided which force field had a higher level of stiffness. In addition to the load force feedback, when interacting with the standard force field (which had a constant stiffness value of 85 N/m), the skin of the thumb and index finger was stretched using a skinstretch device [Fig. 1]. The stiffness of the comparison force field was chosen from three stiffness values (40, 85, and 130 N/m). The magnitude of both the stretch and the load force was proportional to the penetration depth into the virtual elastic force field. The gain of the skin-stretch stimulation (i.e. the amount of skin-stretch relative to the penetration depth) was chosen from four possible values (0, 33, 66, and 100 mm/m). In each trial, participants made eight discrete movements into each of the two force fields. In some of the trials, either the second or the seventh probing movements into the standard force field, were stretchcatch probes. In these stretch-catch probes, we maintained the load force but omitted the skin-stretch. The purpose of the stretch-catch probes was to allow us to investigate the predictive component of grip force control. To identify the predictive component, which develops with repeated stretch stimulation, we compared between the grip force that was applied throughout the second and the seventh stretch-catch probe in a given trial.”

8) The labelling of the axes is sometimes unclear e.g. Fig 2A - the 'probability of responding comparison' is not well defined.

We changed the label to:

“p (answer comparison>standard)”

9) Authors claim that increasing the skin-stretch gain causes more grip force peaks to appear. This is not evident from the presented figure (Figure 3). How was it quantified? By visual inspection?

Thank you for pointing out this lack of clarity. Indeed, this observation was qualitative rather than quantitative, and was supported by a visual examination. It was impossible to quantify this effect because we suspected that in the free exploration probing, the peaks that belonged to subsequent probing movements were fused. This was one of the observations that led to the design of the more structured experiment (now Experiment 1), with sufficient separation time between subsequent peaks and the use of stretch-catch probes. We revised the relevant paragraph to clarify this point:

“Fig. 6 presents examples of grip force, load force, and tactor displacement trajectories of a participant from the free exploration group (Group 2). As in Experiment 1, a visual examination of the trajectories revealed a non-uniform peak pattern that appeared predominantly in trials in which skin-stretch was applied. For example, the grip force signals in Fig. 6(c) and 6(d) show that increasing the tactor displacement gain caused several grip force peaks around the load force peaks. In addition, the grip force trajectories give the impression of the presence of phase shifts between grip force responses and the load force due to added tactor displacement. However, we suggest that in reality participants’ grip force in Experiments 1 and 2 were similar. In Experiment 1, participants made discrete probing movements into the force field, leading to the clear grip force trajectories seen in Fig. 2. In Experiment 2, however, participants used free exploration and the probing movements lacked the separation that existed in Experiment 1. In Experiment 1, we observed two peaks in the grip force trajectories when high tactor displacement gains were applied. We hypothesized that this pattern was due to the reactive grip force component and an artifact specific to our skin-stretch device (see SF for more information). We therefore posit that the free exploration used in Experiment 2 caused the second peak of each probing to fuse with the first peak of the subsequent probing.”

10) The explanation of how the reactive component of grip force control was isolated in Experiment 2 should be made more clearly.

We revised the explanation of how the reactive component was isolated. In addition, following restructuring and focusing of the paper, we moved this entire analysis and interpretation of the reactive component to the Supplementary file. The new explanation of the analysis is as follows:

“First, we identified the onset and end of the second and seventh probing movements by finding the first and last sample in the second and seventh interactions with the force field. Next, we isolated each trajectory from 50 samples before the onset and 50 samples after the end of the interaction. This measure was taken to ensure that we capture all the grip force data. We then divided each grip force trajectory by the peak load force in the same probe, and time-normalized and aligned each trajectory such that 0 and 1 were the onset and end of the contact with the load force. The next step was to separate the trials according to the tactor displacement gain and average the grip force trajectories from the second and seventh probing movements (from both regular and stretch-catch probe trials). To eliminate variability due to baseline grip force level, we subtracted the onset grip force value from each averaged trajectory. Finally, we subtracted the average stretch-catch probes grip force trajectory from the average normal probes grip force trajectory. The remaining signal was the reactive effect of the stretch stimulus on grip force from the early and the late exposure to each tactor displacement gain.”

11) Why do we only see the phase differences between grip force and load force in Experiment 1 (Figure 3), but not in Experiment 2 (Figure 5)? Similarly, why is the 'irregular pattern of decrease accompanied by increase' only apparent in Experiment 2? Weren't those experiments performed in the exact same way, except for the controlled number of probing movements and presence of the stretch-catch trials?

We believe that these two observations (the apparent phase shift and the irregular pattern of decrease accompanied by an increase) are in fact both a result of the same phenomenon. The lower panels in Figure 2 depict a double peaked grip force pattern in the cases of applied skinstretch. We believe that in Fig. 6, what seems as an apparent phase shift is in fact the same picture as in Fig. 2, but the second peak of each probing is fused with the first peak of the subsequent probing. We revised the text describing these phenomena such that the link between them is clearer. The entire Supplementary file deals with the reactive and control experiments, and specifically, the following paragraph discusses the double peaked pattern:

“Fig. 6 presents examples of grip force, load force, and tactor displacement trajectories of a participant from the free exploration group (Group 2). As in Experiment 1, a visual examination of the trajectories revealed a non-uniform peak pattern that appeared predominantly in trials in which skin-stretch was applied. For example, the grip force signals in Fig. 6(c) and 6(d) show that increasing the tactor displacement gain caused several grip force peaks around the load force peaks. In addition, the grip force trajectories give the impression of the presence of phase shifts between grip force responses and the load force due to added tactor displacement. However, we suggest that in reality participants’ grip force in Experiments 1 and 2 were similar. In Experiment 1, participants made discrete probing movements into the force field, leading to the clear grip force trajectories seen in Fig. 2. In Experiment 2, however, participants used free exploration and the probing movements lacked the separation that existed in Experiment 1. In Experiment 1, we observed two peaks in the grip force trajectories when high tactor displacement gains were applied. We hypothesized that this pattern was due to the reactive grip force component and an artifact specific to our skin-stretch device (see SF for more information). We therefore posit that the free exploration used in Experiment 2 caused the second peak of each probing to fuse with the first peak of the subsequent probing.”

12) The quantitative analysis of the peak grip force to peak load force ratio is very problematic in this paper because the tactile stimulation generates major time shifts between Grip Force and Load Force (and multiple peaks for the Grip Force). Therefore, the interpretation of this ratio is very hard at this stage. The reported effects could be due to the predictive feedforward component or to the reactive feedback component or to their interaction. This is why the panel b of Figure 6, for instance, is probably misleading, because there is an asynchrony between Grip Force and Load force in the data (see also comments below). The authors overlook this problem of asynchrony in the paper (see Discussion section: this major issue of asynchrony and multiple peaks in the Grip Force response is ignored).

This is a very important point.

In Experiment 1 (original Experiment 2) we conducted the peak grip force-peak load force ratio analysis only on stretch catch probes (probes without artificial skin-stretch) and therefore we did not have the problems of time shift or multiple peaks. This analysis tested only the effect of the stretch in the preceding probing movements on the predictive component of grip force control in a given stretch-catch probing movement. We highlighted this point in our report of the results:

“To quantify these effects, we analyzed the grip force-load force ratio (Fig. 3). The next analyses refer only to the predictive component of grip force control.”

And in the Materials and methods section:

“To dissociate the contribution of the feedforward grip force adjustments, we incorporated stretch-catch probes where we maintained the load force but unexpectedly omitted the skinstretch (Hermsdörfer and Blankenfeld, 2008). In these probing movements, participants expected to feel the skin-stretch, as they did in the previous probing movement into the same force field. They therefore predictively adjusted their grip force in anticipation of this stimulus. As it was a stretch-catch probe, there was no skin-stretch to apply a reactive grip force to.”

In Experiment 2 (original Experiment 1) we did this analysis on regular probes and therefore we indeed had to deal with time shift and multiple peaks. It is important to note that we did the analysis of peaks ratio in Experiment 2 only to test if we can reproduce some of the results of Experiment 1 in a more natural task. We clarified that the results of this analysis include the contribution of both the predictive and reactive responses:

"Unlike Experiment 1, in which we separated the predictive component into the baseline grip force and the modulation, and analyzed each individually, in this experiment we analyzed the overall grip force. The overall grip force is comprised of a predictive and a reactive component.”

We also highlight the relation between the patterns that were observed in the two experiments:

“Fig. 6 presents examples of grip force, load force, and tactor displacement trajectories of a participant from the free exploration group (Group 2). As in Experiment 1, a visual examination of the trajectories revealed a non-uniform peak pattern that appeared predominantly in trials in which skin-stretch was applied. For example, the grip force signals in Fig. 6(c) and 6(d) show that increasing the tactor displacement gain caused several grip force peaks around the load force peaks. In addition, the grip force trajectories give the impression of the presence of phase shifts between grip force responses and the load force due to added tactor displacement. However, we suggest that in reality participants’ grip force in Experiments 1 and 2 were similar. In Experiment 1, participants made discrete probing movements into the force field, leading to the clear grip force trajectories seen in Fig. 2. In Experiment 2, however, participants used free exploration and the probing movements lacked the separation that existed in Experiment 1. In Experiment 1, we observed two peaks in the grip force trajectories when high tactor displacement gains were applied. We hypothesized that this pattern was due to the reactive grip force component and an artifact specific to our skin-stretch device (see SF for more information). We therefore posit that the free exploration used in Experiment 2 caused the second peak of each probing to fuse with the first peak of the subsequent probing.”

Technically, to deal the problems of time shift and multiple peaks we used the MATLAB function findpeaks to identify the grip force and load force peaks and then manually corrected by visual examination. The visual examination enables us to make sure we analyzed only grip force peaks which preceded or were synchronized with the load force peaks. This was clarified in the Materials and methods section:

“To reproduce the results of the change in the predictive grip force between early and late interactions with the elastic force field (observed in Experiment 1) we computed the peak grip force-peak load force ratio in Group 2 of Experiment 2. Unlike in Experiment 1, in which we only analyzed the grip force-load force trajectories of the stretch-catch probes, in Experiment 2, we examined the effect of the additional skin-stretch on the overall grip force trajectories. The grip force trajectories had a non-uniform peaked pattern that appeared predominantly in trials in which skin-stretch was applied. Therefore, we used the MATLAB function findpeaks to identify the grip force and load force peaks, and then manually corrected them by visual examination.”

13) The last panel of Figure 5 is a very important result from this study: the 7th stretch-catch probe shows that the feedforward predictive component of the Grip Force (in anticipation of a skin stretch that does not occur) is characterized by a single peak (and is larger than if no skin stretch is expected). It also shows that the reactive tactile feedback response reduces the Grip Force response (see double peaks in other trials, same panel). This reduction of the Grip Force was not expected and the authors should better convince the reader that it is not an artefact of their stimulus. I suggest for instance to test the response to the skin stretching when maintaining a STABLE Grip Force, at different typical values, within the Grip Force range tested in the study. Is the Grip Force response also a reduction in this case?

Thank you for this suggestion. Indeed, we agree with both points made by the reviewer, and we clarified the conclusion about the predictive component of the grip force.

“It is also evident from Fig. 2 that in the event of large tactor displacement gains (66 and 100 mm/m), repeated interactions with the force field led to an increase in the predictive grip force control. The grip force trajectories in the stretch-catch probes in panels (b)-(d) show that the feedforward predictive component of the grip force (in anticipation of a skin-stretch stimulus that does not occur) is characterized by a single peak. Moreover, in the seventh probing movement, this peak grip force is larger than it would be if no skin-stretch was expected (compare the seventh movements of panel (a) and (d)).”

Our original intent was to understand both the predictive and the reactive responses, and indeed, we were also surprised by the reduction in the reactive grip force component. We added the proposed experiment of quantifying the response to stretch while maintaining a constant target grip force at two typical baseline values from our original experiment, 1.2N and 1.6N. The methods and results are described in the Supplementary file, under “Reactive Experiment 2”, and the reactive grip force trajectories of the participants in this stable maintenance task is consistent with the reactive trajectory that we isolated from Experiment 1.

“As in the active experiment, we observed a pattern of a decrease in the grip force followed by an increase toward the end of the stretch. Additionally, we found a similar dependence on the tactor displacement gain – both the decrease and the subsequent increase were greater for larger tactor displacement gains. In addition, a larger target grip force led to a bigger decrease (rm-General Linear Model, main effect of ‘gain’: *F*_(1,9)_ = 18.54, *p* = 0.0001, main effect of ‘target grip force’: *F*_(1,9)_ = 8.54, *p* = 0.0053). Both reactive experiments taken together suggest that the pattern of the reactive response was not specific to the control of grip force during probing movements, but rather a more general response to the tactile stimulation that we applied in this study.”

However, even after these new observations, we were still unsure that this is not an artifact of our stimulation. Therefore, we also added the control experiments (Control Experiments 1 and 2), which revealed qualitatively similar patterns of grip force trajectory changes, but that were smaller by factors of four (Control Experiment 1) and eight (Control Experiment 2). With these results, we concluded that:

“It is important to note that the reactive grip force response we report here may be a specific response to our skin-stretch device, and not a general response to any artificial tactile stimulation. In addition, these results should be considered with caution because of the resemblance between the reactive grip force trajectories and the trajectories of the artifact of tactor movement, which we characterized in the two control experiments. However, the participants’ reactive response was much greater than the artifact, suggesting that the artifact is not a plausible explanation for the entire response.”

Finally, because of this ambiguity and to better focus the message, we focus the message of the main paper on the predictive control of grip force in anticipation of the stretch, the effect of the stretch on perception, and the link between the two. Therefore, we moved the entire story of the reactive grip force control to the Supplementary file. If the reviewers and the editor are of the opinion that the parts describing Reactive Experiments 1 and 2, and their conclusions, better fit within the main text, we will be happy to make this change.

14) The authors mention in the discussion section that: "The reactive grip force pattern was very surprising". Indeed, they expected an increase in Grip Force instead of a decrease in reaction to the tactile stimulus. The authors should consider the possibility that the tactile reactive feedback (reduction of Grip Force) might be due to a general mechanism that could be some kind of response to a nociceptive input when there is a mismatch between what you expect from prediction and what you feel at your fingertips (think about a bug that you hold between your fingers and that tries to escape). This response could be specific to the situation when the Grip Force varies (active movement) and not during static holding (see control experiment suggested above) or could alternatively be present in both conditions.

We agree with the reviewers and added the experiment to address this point. Please refer to our response to the previous comment. We also added two potential explanations to this response:

“We do not have a conclusive explanation as to why the reaction to skin-stretch would be an initial release of grip. One possibility is that the release was the result of an unpleasant or painful interaction, however none of our participants indicated any discomfort, nor did they report any when asked about their experience at the end of the experiment. Another possibility is the attempt to reduce the contact impedance between the fingers and the device due to possible contact instability (White et al., 2018).”

In addition, we highlight that:

“It is important to note that the reactive grip force response we report here may be a specific response to our skin-stretch device, and not a general response to any artificial tactile stimulation.”

15) The authors also mention in the discussion section: "This prompted our initial prediction that the reactive response would be an increase in grip force 50-90 msec after the stimulation". This prediction is not correct on the basis of existing literature (Dimitriou et al., 2012; Pruszynski et al., 2016, Crevecoeur et al., 2017). Indeed, the measured EMG response to a tactile stimulus is around 90ms. Therefore, a change in Grip Force is not expected before 120 to 150ms, which is compatible with the reported data for the peak of decrease: "The peak of the decrease was roughly in the middle of the probe, close to the peak of the stretch and the load force, and about 150 msec from the onset of the stretch.".

Thanks for pointing this mistake to our attention. We corrected it and revised the paragraph.

“The reactive grip force pattern was very surprising. At the beginning of an interaction, before an internal representation was formed, a tactile stimulus is similar to any mechanical perturbation. It is well-established in the literature that during unexpected load force perturbations, the grip force is automatically adjusted (Cole and Abbs, 1988; Johansson and Cole, 1992; Cole and Johansson, 1993). Recordings from tactile afferents during trapezoidal load force perturbations to the digit with different rates of loading (Macefield, Häger-Ross and Johansson, 1996) revealed that the mean firing rate was scaled by the slope of the loading ramp. This prompted our initial prediction that the reactive response would be an increase in grip force, which would that increases linearly with increasing tactor displacement gain, and would appear more than 90 msec after stimulation (Dimitriou, Franklin and Wolpert, 2011; Pruszynski, Johansson and Flanagan, 2016; Crevecoeur et al., 2016, 2017). Instead, in response to the artificial stretch, participants first strongly decreased their grip force and only then increased it, resulting in an overall moderate increase that depended linearly on the tactor displacement gain. The peak of the decrease was roughly in the middle of the probe, close to the peak of the stretch and the load force, and about 150 msec from the onset of the stretch. The peak of the increase was after the stretch, and load force returned to zero about 300 msec from the onset of the stretch.”

16) What is puzzling in the data is that the subjects are exposed to a repetition of tactile stimuli but lack adapting their motor response efficiently. Indeed, they cannot inhibit the reduction in Grip Force due to the tactile feedback and consequently show a local minimum of Grip Force when the Load Force is maximum. This is clearly a very bad strategy that is repeated by the subjects who fail to adapt in an efficient way (they react to the increased uncertainty due to the tactile stimulus by boosting the feedforward component). It can only be partially compensated by increasing the feedforward predictive Grip Force component (to compensate for the expected and inefficient reduction of Grip Force after the tactile stimulus) (subsection “Skin-stretch increased the feedforward component of grip force”).

This is an interesting idea, but it is important to note that we interpret the reactive grip force results with caution because the reactive response may be specific to our device and needs to be further investigated.

17) In Experiment 1, the subjects performed several probing movements for each stimulus. When there was a tactile stimulus, they adapted their strategy across repetitions (see Figure 3 last panel and Figure 4B for a comparison between first and last ratio). Therefore, it is not clear what induced the change in PSE. Is it the tactile stimulus or is it the build-up of increased feedforward Grip Force across repetitions with the tactile stimulus? A control experiment involving a test with a single probing movement is necessary from my point of view. It would allow disambiguating the role of the tactile stimulus versus feedforward component in assessing stiffness. An alternative idea would be to test perception of stiffness specifically during catch trials (it remains to be determined how this could be done in a protocol). It would also allow separating the feedforward component (that is increased in catch trials) from the feedback reactive response to the tactile stimulus.

Thank you for the idea for this new control experiment. We added the new control experiment (Experiment 2, Group1), in which participants could touch each force field only once, and compared it to a group which used free exploration (Experiment 2, Group 2). We found that the bias in stiffness perception (as quantified by the PSE) and the discrimination sensitivity (as quantified by the JND) did not depend on the number of probing movements. Therefore, we concluded that the perceptual illusion is formed immediately, and does not depend on the buildup of the internal representation that is used for the predictive control of grip force.

18) In the discussion section the authors mention that: "another possible explanation for the increased grip force is that the motor system of the participants could have interpreted the additional tactile stimuli as indication of a more slippery contact surface, which would have thus been entirely unrelated to the perceptual illusion of a harder spring." This is a very interesting idea. I encourage the authors to add analyses related to the correlation between the Grip Force and Load Force (phase diagram) during the first open-loop phase of the probing movements (Johansson and Westling, 1984).

Thank you for this positive feedback. In our original submission, we analyzed the grip force load force phase diagram and fitted a regression line to these trajectories (similar to the analysis in (Flanagan and Wing, 1990; Leib et al., 2018, 2015)). This analysis enables us to differentiate between the two components of the predictive grip force control: (1) the baseline, which provides a safety margin against slippage and does not use a specific representation of the force field, and (2) the modulation in anticipation of the load force, which requires an internal representation of the load force and the slipperiness. We clarified this analysis in the Introduction and the results section:

“Feedforward control is used to adjust our grip force in accordance with the expected slipperiness and weight (or load force) of the object (Flanagan and Wing, 1990, 1997; Flanagan et al., 2003; Danion and Sarlegna, 2007; Danion, Descoins and Bootsma, 2009; Leib, Karniel and Nisky, 2015; Leib, Rubin and Nisky, 2018; White et al., 2018). Baseline grip force is maintained to create a safety margin, and is increased when one experiences uncertainty regarding the load force (Gibo, Bastian and Okamura, 2013; Hadjiosif and Smith, 2015). Predictive modulation is calculated based on an internal representation of the object dynamics, which is learned by the nervous system (Johansson and Cole, 1992; Shadmehr and Mussa-Ivaldi, 1994; Kawato, 1999; Davidson and Wolpert, 2004) and is updated during repeated interactions (Donchin, Francis and Shadmehr, 2003).”

“The intercept of this regression represents the baseline grip force; i.e. the amount of grip force that was applied by the participants when no external load force was applied by the haptic device. The slope of the regression represents the modulation of the grip force that anticipates the load force. In typical manipulation tasks of grasping and lifting, the slope is linked to the “slip ratio” of an object (Flanagan and Wing, 1990), and when interacting with elastic objects, to a combined effect of the slip ratio and the internal representation of the load force (Flanagan and Wing, 1990; Leib et al., 2018, 2015).”

The correlation between the derivative of GF and LF should also be very informative. This would allow assessing the feedforward predictive component of Grip Force even during standard probing movements (Experiment 1) and not only during catch trials (Experiment 2).

Indeed, in many of the classical and recent studies of the modulation of grip force with load force the relationship between GF rate and LF rate is informative, see e.g. (White et al., 2011). However, these studies focus on lifting tasks where there is a clear onset of load force at the lift-off inception. When the load force is elastic, like in our case, a correlation between grip force and load force is also used (Descoins et al., 2006), but so does the analysis of regression in the grip force load force plane (Flanagan and Wing, 1990). We tried to perform an analysis of GF-LF rate on the standard probing movements, but in these movements, it was impossible to associate the specific local peaks of the grip force rate, tactor movement rate, and load force rate due to artifacts that are related to the movement of the tactor during these probing movements, so eventually, this analysis was not informative. Therefore, we chose to focus on the information that can be reliably inferred from the grip force load force regression during the stretch-catch trials.

19) In the discussion section, the authors mention that: "Future studies, probably with other designs of tactile stimulation devices, are needed to pinpoint the origin of this surprising reactive pattern." Given that the same group already reported the finding of the change in the perception of stiffness following a tactile stimulus (papers by Quek et al.), I wonder whether better characterizing and understanding the nature of the reactive pattern is not necessary to make the present study truly new, interesting and innovating.

In this new submission we present novel results about the predictive component of grip force control and the timescale of forming stiffness perception during subsequent probing movements. We remain careful about the interpretation of the meaning of the reactive component, and therefore, following restructuring and focusing of the paper, we moved the entire analysis of the reactive component of grip force control to the Supplementary file.

[Editors' note: further revisions were suggested prior to acceptance, as described below.]

Summary:The authors address a specific question, related to control of the grip force that all of us use with no difficulty when we grasp an object like a paper-cup of coffee. This action requires accurate modulation of the grip, at an accurate and well-timed force, to prevent the cup slippage, (spilling the coffee), and at the same time avoiding crash the paper cup. Investigation of this question is related to many aspects of action-perception behaviors and beyond. The major challenge in all cases is not just the action itself but in the prediction of the future, given the present (and past) concepts and actions. Therefore, the potential projection of this study to more general science fields is part the general significance of this study.Here, the authors investigated the effect of stretching the skin of the fingertip while performing movements in a force field. The force field simulated the stiffness of a spring in a virtual environment. The goal was to determine how the tactile and kinesthetic feedbacks are combined and influence both stiffness perception and grip force control. The authors also used catch trials to try and separate the three components of the Grip Force response: predictive feedforward baseline, predictive feedforward modulation and reactive feedback. The authors address an interesting and very important question related to the role of tactile inputs on perception and motor control. Experiment 2 based on stretch-catch probes reports very robust and original data showing a negative reactive response of Grip Force to skin stretching. The authors also performed several control experiments to better assess the limitations of their finger stimulation device. The results on the influence of tactile stimulation on the perception of stiffness are very robust and confirm earlier reports from the same group. The demonstration that higher levels of artificial skin-stretch increased the predictive grip force modulation in anticipation of the load force and the finding that the increase developed throughout each trial are important. Thus, it is also important to improve and quantify the temporal aspects of the measurements.The manuscript has been greatly improved but there are some remaining issues that need to be addressed before acceptance.

Thank you very much for having considered our work and inviting us to submit the revised manuscript. We have carefully evaluated the recommended revisions and made the necessary changes: we added the requested analyses, conducted new control experiments, and thoroughly edited all the sections of the paper. We believe that our paper has been considerably improved as a result of these revisions.

1) We ask that you provide a better characterization of the evolution of grip force over time.

We adopted the analyses in Kuling et al., 2019 with very minor modifications, as specified in the methods section. These new analyses enable us to characterize the evolution of the predictive (baseline and modulation) grip force control over repeated probing movements within each trial. The results of this analyses corroborated our original assertion that: “artificial skin-stretch increased the predictive grip force by increasing both the baseline grip force and the modulation in anticipation of the predicted load force”.

Importantly, with these new analyses we can now assert with greater confidence what we could only hypothesize in the previous version of our submission – that:

“the timescales of the evolution of these two components were different – the baseline grip force increased immediately together with the perceptual illusion, and the modulation evolved gradually.”

We added new subsections describing these new analyses and their results:

Subsection: “Baseline grip force and prediction of the intended peak grip force in all the probing movements in each trial”.

and subsection “The evolution of the predictive grip force over repeated probing movements”.

We also added two new figures: Figure 5 and Figure 6.

We modified the conclusions throughout the text such that the gradual nature of the development of the predictive modulation is highlighted, and modified the title to: “Stretching the Skin Immediately Enhances Perceived Stiffness and Gradually Enhances the Predictive Control of Grip Force”

2) The results on the influence of tactile stimulation on the perception of stiffness are very robust and confirm earlier reports from the same group. In contrast, the results on the influence of tactile stimulation on Grip Force control remain very difficult to interpret despite the additional control experiments performed by the authors. This could be partly improved by conducting additional analyses as suggested below.a) In their analyses, the authors try to separate the baseline and modulation of Grip Force (Results section): "To separate between the effect on each of these two components, we performed a linear regression analysis in the grip force-load force plane." This is illustrated in Figure 4 with phase plots of Grip Force versus Load Force. However, this figure shows that the relationship between Grip Force and Load Force is not very linear. This is also clear in Figure 2 because the Load Force profile is very stereotyped and short in time in comparison with the Grip Force profile that starts increasing well before the increase in Load Force in anticipation of the contact with the virtual object. This observation is classical in all studies investigating collisions between (virtual) objects. In that case, it is much more appropriate to measure the baseline Grip Force as the average Grip Force level between two consecutive trials (when Grip Force does not vary much over time, see Figure 2D). It is also more appropriate to measure the modulation when both Grip Force and Load Force vary quickly during the early part of the contact. This kind of analysis was recently used in the recent paper of Kuling et al., (2019). In the same paper, these authors also propose a method to compute an estimate of the predictive component of the feedforward Grip Force on the basis of a regression analysis taking into account Grip Force at contact (with the virtual object) and the derivative of Grip Force at contact. This method was validated on catch trials when there was no contact with a virtual object. This method is interesting in the context of the present study because it would allow further investigating the gradual build-up of Grip Force feedforward component across the repetition of trials with tactor displacement. The following reference could also be cited (Grip Force during collisions in a virtual reality setup): White et al., (2011).

Thank you for these suggestions, which we fully took on board.

We added a new subsection “The evolution of the predictive grip force over repeated probing movements” to the results of Experiment 1:

This section quantifies the evolution of both the baseline grip force and the intended peak grip force between consecutive probes. The intended peak grip force was predicted using a multivariate regression with the grip force at contact and its rate of change as the independent variables. Additionally, we quantified the development of the grip force modulation, which was defined as the difference between the intended peak grip force and the grip force at contact, as well as the grip force at contact over subsequent trials.

The new analyses not only corroborated our original conclusions but also allowed us to determine that the development of the predictive modulation of grip force in anticipation of load force was gradual. The following conclusion was clarified in the Discussion section:

“Our results showed that artificial skin-stretch increased the predictive grip force by increasing both the baseline grip force and the modulation in anticipation of the predicted load force. In addition, we determined that the size of this effect was linear with the amount of stretch. Interestingly, the timescales of the evolution of these two components were different – the baseline grip force increased immediately together with the perceptual illusion, and the modulation evolved gradually.”

b) In the Materials and methods section and throughout the paper, the authors acknowledge that their "device only measures a downscaled version of the grip force". Of course, this introduces a major limitation in the interpretation of the experimental data related to the measure of Grip Force. At this stage, there is no quantification of the relationship between actual Grip Force and the measured downscaled version of Grip Force. We do not even know if the relationship will be linear or non-linear given the complex mechanics of the object / finger pad interaction (pressure is not constant in the contact area). In their paper, the authors tried to address this important issue but in a way that is not satisfactory given the importance of Grip Force measurement. I strongly recommend performing the following very simple control experiment to clarify this issue. The authors should simply disconnect their manipulandum device from the haptic robot and put it flat on an extra (1D) force sensor. They should then instruct a few human subjects to exert a pure normal (vertical) force on the device over the range of recorded Grip Force in the experiments. They should then compare the Grip Force measure with the device with the true total normal force measured by the additional force sensor to assess the "downscaling" of the measured Grip Force with the device. This could be done with or without tactor displacement for a more thorough control measurement.

We performed the new proposed experiment and verified that indeed the mapping between the actual grip force and the downscaled measurement of the grip force is close to linear. The movement of the tactors leads to a reactive response and an artifact [as characterized in Supplementary file section (1)]. Therefore, the majority of the grip force analyses in our paper were performed using portions of the interaction without tactor movement. Due to these reasons, this experiment was conducted without tactor movement.

We described this experiment and its results in the Supplementary file subsection (2) “The mapping between the actual grip force and the downscaled measurement of the grip force”.

1) The experiment does not show directly any "internal representations" of the well-demonstrated illusion. In general, the term It is an over-used, and means very little. Of course, the authors can see it their way, and I do not demand the elimination of terms. I counted 28 times (twice in the Abstract!) I do recommend to reduce this number or abandoning this term, leaving the issue of neuronal basis to speculation/interpretation alone. For example, the text of the authors:"This suggests that the perceptual illusion is a result of a high-level cognitive process, independent of the gradual development of the internal representation that causes an increase in the predictive grip force control.Could be simpler... like:"This suggests that the perceptual illusion is a result of an adaptive predictive process, independent of the gradual development of grip force control during the action".

We reduced significantly the number of uses of this term and rephrased many of our statements to be less speculative.

2) The results on the influence of tactile stimulation on the perception of stiffness are very robust and confirm earlier reports from the same group. In contrast, the results on the influence of tactile stimulation on Grip Force control remain very difficult to interpret despite the additional control experiments performed by the authors. This could be partly improved by conducting additional analyses as suggested below.

We added the suggested analyses.

3) The authors report (Results section) that "participants applied more grip force per amount of kinesthetic load force when interacting with force fields with higher levels of tactor displacement gain." This observation makes sense because the skin stretch induces a large decrease of Grip Force around the peak of Load Force. It is particularly clear for large tactor displacement (100 mm/m, Figure 2D) where there is a clear dip in Grip Force around the peak of Load Force (see comparison between trial 6 and 7 in Figure 2D: difference in Grip Force of 2 N around the peak Load Force). Therefore, this dip in Grip Force around the peak of Load Force increases the risk of slippage which is compensated by an increase of baseline and feedforward Grip Force components (see comments below).

We added this additional interpretation to the discussion section:

“Moreover, we observed a reactive reduction of the grip force around the peak load force in response to the stretch stimulation (which we interpret with caution - see SF). This reduction increases the risk of slippage which could be compensated for by increasing both components of the predictive grip force (i.e., the baseline and the modulation) that are entirely unrelated to the mechanical properties of the interface with the finger.”

4) Very interestingly, the authors mention in the same paragraph (Results section) that "This trend is significantly higher in the seventh probing compared to the second". This is a very important observation because it shows that the increase in baseline and feedforward Grip Force components is clearly decoupled from the perception of increased stiffness that is shown in the manuscript to take place after a single trial as demonstrated in Experiment 2 and Figure 5 (subsection “Skin-stretch caused perceptual overestimation of stiffness”): "We can therefore conclude that the perceptual illusion was formed immediately, after a single probing movement into the force field, and before the internal representation that we identified in Experiment 1 was formed. "

We highlighted this observation, which was characterized still further with the new analyses proposed by the reviewers.

In the Results section:

“To summarize, we found that following exposure to high levels of artificial skin-stretch participants immediately increased their safety margin and maintained it even after many interactions. They also increased the predictive grip force modulation in anticipation of the load force. This increase developed gradually with subsequent probing interactions with the same elastic force field.”

In the Discussion section:

“Our results suggest that artificial skin-stretch also affected the prediction of the load force applied by the object, and consequentially led to a gradual increase in the predictive modulation of grip force in anticipation of the load force. Additionally, we discovered that the perceptual illusion was formed immediately upon first contact with the object, and did not depend on the slower development of the predictive control of grip force.”

“Our results showed that artificial skin-stretch increased the predictive grip force by increasing both the baseline grip force and the modulation in anticipation of the predicted load force. In addition, we determined that the size of this effect was linear with the amount of stretch. Interestingly, the timescales of the evolution of these two components were different – the baseline grip force increased immediately together with the perceptual illusion, and the modulation evolved gradually.”

5) Results section: "This suggests that following repeated exposure to the artificial skin-stretch stimulation, the participants either increased their internal representation of the expected force (likely due to increase in the representation of the stiffness of the field) or of the slipperiness of the device." Again, see comments above: the fact that the perception of stiffness occurs much faster and that the measured effects on catch trials 2 and 7 (Figure 4F) are very different seems to demonstrate that the reported effect is not due to gradual increase in the internal representation of the expected force but more likely to a slower adaptation of Grip Force potentially due to an increased risk of slippage ("reactive" dip in Grip Force).

We agree that this is a possible interpretation and added it to the Discussion section:

“Moreover, we observed a reactive reduction of the grip force around the peak load force in response to the stretch stimulation (which we interpret with caution - see SF). This reduction increases the risk of slippage which could be compensated for by increasing both components of the predictive grip force (i.e., the baseline and the modulation) that are entirely unrelated to the mechanical properties of the interface with the finger.”

6) One key experiment for the interpretation of the Grip Force data of the current study is the Reactive Experiment 2 reported in the Supplementary File. I believe that it is crucial to fully exploit this data set to better understand the effect of tactor displacement on the "reflex" Grip Force response reported in the main experiment. I even think that a better controlled experiment based on the analyses of this Reactive Experiment should be published as an independent study before publishing the current set of experiments that cannot be satisfactorily interpreted in terms of Grip Force control at this stage. I am convinced that it could be a major contribution to our understanding of the very important question related to the role of proprioceptive versus tactile feedbacks in motor control (Crevecoeur et al., 2017). The device used by the authors allows investigating the impact of a pure tactile stimulation on the motor response (Grip Force reaction). I recommend redoing this reactive experiment and recording EMG signals together with the measurement of the Grip Force reaction. This will allow measuring the delay of the motor response and better characterizing the timing of the reaction which is crucial to understand the results of the current study (the delay seems very short in Figure 10: shorter than 20ms?).

Thank you for this very interesting suggestion. Going forward with our research program, we indeed intend to measure EMG in experiments conducted with our skin-stretch device. We are confident that once the setup is established and validated in our lab, this direction will lead to new discoveries. However, we feel that adding such an experiment would be unrealistic in the review timeline as it would require developing a completely new set of tools and analyses. We therefore look forward to investigating this topic in the continuation of our research.

7) Figure 7 shows the ratio between peak Grip Force and peak Load Force. It is not clear what this analysis means because in Figure 6 the peaks of Grip Force and Load Force are not synchronized! In the current version of the manuscript (and reported analyses) the interpretation of Grip Force data is very weak (see also Materials and methods section).

Indeed, this analysis of grip force in Experiment 2 is potentially affected by interfering reactive responses and potential artifacts of tactor movement [see Supplementary file section (1)]. Our major conclusions about the control of grip force in this manuscript are based on Experiment 1 (original Figure 2, Figure 3, Figure 4, and new Figure 5 and Figure 6). We chose to keep this analysis because it corroborates our conclusions from Experiment 1.

If you prefer not to include this analysis in the paper, it’s all right by us to remove it.

8) In the Discussion section, the authors mention that "in our case, increasing the grip force reached the perceptual illusion over repeated interactions". I am not sure that the interpretation of the authors is correct (see also comments above) because there is an illusion in terms of stiffness but not in terms of tactile sensation: there is an actual stretch of the skin, a "reflex" dip in Grip Force, and therefore a sensation of partial slip or strain in the fingertip. Moreover, the time course of the two phenomena is very different: immediate illusion in comparison with progressive build-up of Grip Force anticipation, pointing to very likely different mechanisms.

This phrasing was indeed confusing; the reviewer is indeed correct in the interpretation of the timescales. The original sentence was there to highlight the difference between our findings and the process reported in Flanagan and Beltzner, 2000. We rephrased this sentence and it now reads:

“However, the nature of the change in the grip force was different - in (Flanagan and Beltzner, 2000) the grip force matched the perception in the first grasping, and was then adjusted to the correct weight. In our case, on the other hand, the predictive modulation of grip force slowly increased over repeated interactions.”